# Distribution Learnability and Robustness[*]

**Shai Ben-David**
University of Waterloo, Vector Institute
shai@uwaterloo.ca

**Alex Bie**
University of Waterloo
yabie@uwaterloo.ca

**Gautam Kamath**
University of Waterloo, Vector Institute
g@csail.mit.edu

**Tosca Lechner**
University of Waterloo
tlechner@uwaterloo.ca

## Abstract

We examine the relationship between learnability and robust (or agnostic) learnability for the problem of distribution learning. We show that learnability of a distribution class implies robust learnability with only additive corruption, but not if there may be subtractive corruption. Thus, contrary to other learning settings (e.g., PAC learning of function classes), realizable learnability does not imply agnostic learnability. We also explore related implications in the context of compression schemes and differentially private learnability.

## 1 Introduction

Distribution learning (sometimes called *density estimation*) refers to the following statistical task: given i.i.d. samples from some (unknown) distribution $p$, produce an estimate of $p$. This is one of the most fundamental and well-studied questions in both statistics [DL01] and computer science [Dia16], often equivalent to classic problems of parameter estimation (e.g., mean estimation) in parametric settings. It is easy to see that no learner can meaningfully approximate any given $p$ without having some prior knowledge. The problem then becomes: assuming the sample generating distribution $p$ belongs to a given class of distributions $\mathcal{C}$, and given parameters $\varepsilon, \delta \in (0, 1)$, output some distribution $\hat{p}$ such that with probability at least $1 - \delta$, the statistical distance between $p$ and $\hat{p}$ is at most $\varepsilon$. Specifically, we employ *total variation distance*, the most studied metric in density estimation [DL01, Dia16], using $d_{\mathrm{TV}}(p, q)$ to denote the distance between distributions $p$ and $q$. This case, when $p \in \mathcal{C}$, is often called the *realizable* setting. If, for some particular class $\mathcal{C}$, this is doable with a finite number of samples $n(\varepsilon, \delta)$, then we say the distribution class is $(\varepsilon, \delta)$-*learnable*.[1] A class is *learnable* if it is $(\varepsilon, \delta)$-*learnable* for every $(\varepsilon, \delta) \in (0, 1)^2$. A significant amount of work has focused on proving bounds on $n(\varepsilon, \delta)$ for a number of classes $\mathcal{C}$ of interest – for example, one can consider the class $\mathcal{C}$ of all Gaussian distributions $\mathcal{N}(\mu, \Sigma)$ in some Euclidean space $\mathbb{R}^d$.

However, this framework is restrictive in the sense that it requires the unknown distribution to be *exactly* a member of the class $\mathcal{C}$ of interest. This may not be the case for a variety of possible reasons, including some innocuous and some malicious. As one example, while it is a common modelling assumption to posit that data comes from a Gaussian distribution, Nature rarely samples exactly from Gaussians, we consider this only to be a convenient *approximation*. More generally, the class $\mathcal{C}$ that the learner assumes can be thought of as reflecting some prior knowledge about the task at hand. Such prior knowledge is almost always only an approximation of reality. Alternatively, we may be in an adversarial setting, where a malicious actor has the ability to modify an otherwise well-behaved

---

[*]Authors are listed in alphabetical order.

[1]For the sake of exposition, we defer formal definitions of our learnability notions to Section 1.1.

37th Conference on Neural Information Processing Systems (NeurIPS 2023).

distribution, say by injecting datapoints of their own (known in the machine learning literature as data poisoning attacks [BNL12, SKL17, DKK+19, GTX+20, GFH+21, LKY23]).

More formally, the classic problem of *agnostic* learnability is generally described as follows: given a (known) class of distributions $\mathcal{C}$, and a (finite) set of samples drawn i.i.d. from some (unknown) distribution $p$, find a distribution $\hat{p}$ whose statistical distance from $p$ is not much more than that of the closest member of $\mathcal{C}$. It is not hard to see that this is equivalent to a notion of *robust* learnability, where the distribution $p$ is not viewed as arbitrary, but instead an adversarial corruption of some distribution within $\mathcal{C}$.[2] Given their equivalence, throughout this work, we will use agnostic and robust learnability interchangeably.

The difference between a robust setting and the previous realizable one is that now, instead of assuming $p \in \mathcal{C}$ and asking for an arbitrarily good approximation of $p$, we make no prior assumption about the data-generating distribution and only ask to approximate as well as (or close to) what the best member of some "benchmark" class $\mathcal{C}$ can do.

We address the following question: Assuming a class of distributions is learnable, under which notions of robustness is it guaranteed to be robustly learnable? We focus entirely on information-theoretic learnability, and eschew concerns of computational efficiency. Indeed, our question of interest is so broad that computationally efficient algorithms may be too much to hope for.

We shall consider a few variants of robust learnability. Specifically, we will impose requirements on the nature of the difference between the data-generating distribution $p$ and members of the class $\mathcal{C}$. Obviously, such requirements can only make the task of robust learning easier.

One such model considers *additive robustness*. The underlying distribution is restricted to be a mixture of a distribution $p$ from $\mathcal{C}$, and some 'contaminating' distribution $q$. In this Statistics community, this celebrated model is known as *Huber's contamination model* [Hub64]. Analogously, one can consider *subtractive robustness*. It includes the case where the starting point is a distribution in the class $\mathcal{C}$, but a fraction of the probability mass is removed and samples are drawn from the resulting distribution (after rescaling). These two models are related to adversaries who can add or remove points from a sampled dataset, see discussion at the end of Section 1.2.

A significant line of work focuses on understanding the sample complexity of agnostic distribution learning (see examples and discussion in Section 1.3). Most study restricted classes of distributions, with analyses that are only applicable in certain classes. Some works have found quantitative separations between the different robustness models. For instance, in the specific case of Gaussian mean estimation, [DKK+16, LRV16, DKS17, DKK+18] give strong evidence that efficient algorithms can achieve better error if they must only be additively robust, rather than robust in general. However, such findings are again restricted to specific cases, and say little about the overall relationship between learnability in general and these various robust learning models.

Current results leave open a more comprehensive treatment of robustness in distribution learning. Specifically, what is the relative power of these different robustness models, and what is their impact on which types of distributions are learnable? Are there more generic ways to design robust learning algorithms?

Our two main contributions are the following:

- We give a generic algorithm which converts a learner for a distribution class into a learner for that class which is robust to additive corruptions.

- We show that there exist distribution classes which are learnable, but are no longer learnable after subtractive corruption.

Stated succinctly: we show that learnability implies robust learnability when an adversary can make additive corruptions, but not subtractive corruptions. Other results explore implications related to compression schemes and differentially private learnability.

---

[2]A related notion of robust learnability instead imagines the adversary modifies the *samples* from a distribution in $\mathcal{C}$, rather than the distribution itself. This *adaptive* model is discussed further in Section 1.2.

## 1.1 Definitions of Learnability

In order to more precisely describe our results, we define various notions of learnability. We start with the standard notion of PAC learnability for a distribution class. We get samples from a distribution $p$ belonging to a distribution class $\mathcal{C}$, and the goal is to output a distribution similar to $p$.

**Definition 1.1** (Learnability)**.** *We say that a class $\mathcal{C}$ of probability distributions is* learnable *(or, realizably learnable) if there exists a learner $A$ and a function $n_{\mathcal{C}} : (0,1)^2 \to \mathbb{N}$, such that for every probability distribution $p \in \mathcal{C}$, and every $(\varepsilon, \delta) \in (0,1)^2$, for $n \geq n_{\mathcal{C}}(\varepsilon, \delta)$ the probability over samples $S$ of that size drawn i.i.d. from the distribution $p$ that $d_{\mathrm{TV}}(p, A(S)) \leq \varepsilon$ is at least $1 - \delta$.*

We next introduce the more challenging setting of robust, or agnostic, learning. In this setting, the sampled distribution is within bounded distance to the distribution class $\mathcal{C}$, rather than being in $\mathcal{C}$ itself. For technical reasons, we introduce two closely-related definitions. Roughly speaking, the latter definition assumes the distance from the sampling distribution to $\mathcal{C}$ is fixed, whereas the former (more commonly considered in the agnostic learning literature) doesn't. Note that in many cases, robust algorithms designed with knowledge of the distance $\eta$ to $\mathcal{C}$ can be modified to do without [JOR22].

**Definition 1.2** (Robust learnability)**.**

1. *For $\alpha > 0$, we say that a class $\mathcal{C}$ of probability distributions is $\alpha$-robustly learnable (also referred to as $\alpha$-agnostically learnable) if there exists a learner $A$ and a function $n_{\mathcal{C}} : (0,1)^2 \to \mathbb{N}$, such that for every probability distribution $p$, and $(\varepsilon, \delta) \in (0,1)^2$, for $n \geq n_{\mathcal{C}}(\varepsilon, \delta)$ the probability over samples $S$ of that size drawn i.i.d. from the distribution $p$ that $d_{\mathrm{TV}}(p, A(S)) \leq \alpha \min\{d_{\mathrm{TV}}(p, p') : p' \in \mathcal{C}\} + \varepsilon$ is at least $1 - \delta$.*

   *When $\alpha = 1$ we omit it and say that the class is robustly (or agnostically) learnable.*

2. *For $0 \leq \eta \leq$ and $\alpha > 0$, we say that a class $\mathcal{C}$ of probability distributions is $\eta$-$\alpha$-robustly learnable if there exists a learner $A$ and a function $n_{\mathcal{C}} : (0,1)^2 \to \mathbb{N}$, such that for every probability distribution $p$ such that $\min\{d_{\mathrm{TV}}(p, p') : p' \in \mathcal{C}\} \leq \eta$ and $(\varepsilon, \delta) \in (0,1)^2$, for $n \geq n_{\mathcal{C}}(\varepsilon, \delta)$ the probability over samples $S$ of that size drawn i.i.d. from the distribution $p$ that $d_{\mathrm{TV}}(p, A(S)) \leq \alpha \eta + \varepsilon$ is at least $1 - \delta$.*

Finally, we introduce notions of robust learnability which correspond to only additive or subtractive deviations from the distribution class $\mathcal{C}$. These more stringent requirements than standard (realizable) learnability, but more lenient than $\eta$-$\alpha$-robust learnability: the adversary in that setting can deviate from the distribution class $\mathcal{C}$ with both additive and subtractive modifications simultaneously.

**Definition 1.3** (Additive robust learnability)**.** *Given parameters $0 \leq \eta \leq 1$ and $\alpha > 0$, we say that a class $\mathcal{C}$ of probability distributions is $\eta$-additive $\alpha$-robustly learnable if there exists a learner $A$ and a function $n_{\mathcal{C}} : (0,1)^2 \to \mathbb{N}$, such that for every probability distribution $q$, every $p \in \mathcal{C}$, and $(\varepsilon, \delta) \in (0,1)^2$, for $n \geq n_{\mathcal{C}}(\varepsilon, \delta)$ the probability over samples $S$ of that size drawn i.i.d. from the distribution $\eta q + (1 - \eta)p$, that $d_{\mathrm{TV}}(A(S), p) \leq \alpha \eta + \varepsilon$ is at least $1 - \delta$.*

**Definition 1.4** (Subtractive robust learnability)**.** *Given parameters $0 \leq \eta \leq 1$ and $\alpha > 0$, we say that a class $\mathcal{C}$ of probability distributions is $\eta$-subtractive $\alpha$-robustly learnable if there exists a learner $A$ and a function $n_{\mathcal{C}} : (0,1)^2 \to \mathbb{N}$, such that for every probability distribution $p$ for which there exists a probability distribution $q$ such that $\eta q + (1 - \eta)p \in \mathcal{C}$, and for every $(\varepsilon, \delta) \in (0,1)^2$, for $n \geq n_{\mathcal{C}}(\varepsilon, \delta)$ the probability over samples $S$ of that size drawn i.i.d. from the distribution $p$, that $d_{\mathrm{TV}}(A(S), p) \leq \alpha \eta + \varepsilon$ is at least $1 - \delta$.*

## 1.2 Results and Techniques

We explore how different robustness models affect learnability of distributions, showing strong separations between them. Our first main result shows that learnability implies additively robust learnability.

**Theorem 1.5.** *Any class of probability distributions $\mathcal{Q}$ which is realizably learnable, is also $\eta$-additively 2-robustly learnable for every $\eta \in (0, 1/4)$.*

Note that, since additively robust learnability trivially implies learnability, this shows an *equivalence* between learnability and additively robust learnability.

Our algorithm enumerates over all subsets of the dataset of an appropriate size, such that at least one subset contains no samples from the contaminating distribution. A realizable learner is applied to

each subset, and techniques from hypothesis selection [Yat85, DL96, DL97, DL01] are used to pick the best of the learned distributions. Further details appear in Section 2.

We also note that since our robust learning algorithm enumerates all large subsets of the training dataset, it is *not* computationally efficient. Indeed, for such a broad characterization, this would be too much to ask. Efficient algorithms for robust learnability are an exciting and active field of study, but outside the scope of this work. For further discussion see Section 1.3.

Our other main result shows that a distribution class being learnable does *not* imply that it is subtractive robustly learnable.

**Theorem 1.6.** *For every $\alpha > 0$, there exists a class that is learnable, but not $\eta$-subtractively $\alpha$-robustly learnable for any $0 \leq \eta \leq \frac{1}{16\alpha}$.*

An immediate corollary is that learnability does *not* imply robust (or agnostic) learnability, since this is a more demanding notion than subtractive robust learnability.

Our proof of this theorem proceeds by constructing a class of distributions that is learnable, but classes obtained by subtracting light-weight parts of these distributions are not $\alpha$-robustly learnable with respect to the original learnable class. More concretely, our construction works as follows. We start by considering a distribution class that, by itself, is not learnable with any finite number of samples. We map each distribution in that class to a new distribution, which additionally features a point with non-trivial mass that "encodes" the identity of the distribution, thus creating a new class of distributions which *is* learnable. Subtractive contamination is then able to "erase" this point mass, leaving a learner with sample access only to the original (unlearnable) class. Our construction is inspired by the recent construction of Lechner and Ben-David [LBD23], showing that the learnability of classes of probability distributions cannot be characterized by any notion of combinatorial dimension. For more details, see Section 3.

Thus far, we have only considered additive and subtractive robustness separately. General robustness, where probability mass can be both added *and* removed, is more powerful than either model individually. However, if a class is additive robustly learnable *and* subtractive robustly learnable, is it robustly learnable? Though this is intuitively true, we are not aware of an immediate proof. Using a similar argument as Theorem 1.5, we derive a stronger statement: that subtractively robust learnability implies robust learnability.

**Theorem 1.7.** *If a class $\mathcal{C}$ is $\eta$-subtractive $\alpha$-robustly learnable, then it is also $\eta$-$(2\alpha + 4)$-robustly learnable.*

Adjacent to distribution learning is the notion of *sample compression schemes*. Recent work by Ashtiani, Ben-David, Harvey, Liaw, Mehrabian, and Plan [ABDH+18] expanded notions of sample compression schemes to apply to the task of learning probability distributions. They showed that the existence of such sample compression schemes for a class of distributions imply the learnability of that class. While the existence of sample compression schemes for classification tasks imply the existence of such schemes for robust leaning, the question if similar implication hold for distribution learning schemes was not answered. We strongly refute this statement. We use a construction similar to that of Theorem 1.6, see Section 3.1 for more details.

**Theorem 1.8.** *The existence of compression schemes for a class of probability distributions does not imply the existence of robust compression schemes for that class.*

Finally, a natural question is whether other forms of learnability imply robust learnability. We investigate when *differentially private*[3] (DP) learnability does or does not imply robust learnability. We find that the same results and separations as before hold when the distribution class is learnable under *approximate* differential privacy (i.e., $(\varepsilon, \delta)$-DP), but, perhaps surprisingly, under *pure* differential privacy (i.e., $(\varepsilon, 0)$-DP), private learnability implies robust learnability for all considered adversaries.[4]

**Theorem 1.9** (Informal)**.** *$(\varepsilon, 0)$-DP learnability implies robust $(\varepsilon, 0)$-DP learnability. For any $\delta > 0$, $(\varepsilon, \delta)$-DP learnability implies additively robust learnability, but not subtractively robust learnability.*

---

[3]Differential privacy is a popular and rigorous notion of data privacy. For the definition of differential privacy, see Section 4.

[4]In the context of differential privacy, we diverge slightly from the established notation. Specifically, we align ourselves with common notation in the DP literature, using $\varepsilon$ and $\delta$ for privacy parameters, and use $\alpha$ (in place of $\varepsilon$) and $\beta$ (in place of $\delta$) for accuracy parameters.

For pure DP learnability, we employ an equivalence between learnability under pure differential privacy and packing [BKSW19]. Existence of such a packing in turn implies learnability under both pure differential privacy and with both additive and subtractive contamination. For approximate DP learnability, we note that the corresponding version of Theorem 1.5 automatically holds, since private learnability implies learnability. We show that our construction for Theorem 1.6 is still learnable under approximate differential privacy, and thus the corresponding non-implication holds. See Section 4 for more details.

To summarize the qualitative versions of our findings:

- Learnability and additively robust learnability are equivalent (Theorem 1.5);
- Learnability does not imply subtractively robust learnability (Theorem 1.6);
- Subtractively robust learnability implies robust learnability (Theorem 1.7);
- Pure DP learnability is equivalent to robust pure DP learnability (Theorem 1.9);
- Approximate DP learnability implies additively robust learnability,[5] but not subtractively robust learnability (Theorem 1.9);
- Existence of sample compression schemes does not imply the existence of robust sample compression schemes (Theorem 1.8).

Quantitative versions of these statements can be found in their respective theorems.

**Adaptive Adversaries.** Our definition of robustness allows an adversary to make changes to the underlying distribution. Equivalently, it corresponds to an adversary who can add or remove points from a dataset, but must commit to these modifications *before* the dataset is actually sampled. A stronger[6] adversary would be able to choose which points to add or remove *after* seeing the sampled dataset. Such an adversary is referred to as *adaptive*. Since adaptive adversaries are stronger than the ones we consider, any impossibility result that we show also holds in this settings (e.g., learnability does not imply subtractive robust learnability when the adversary is adaptive). It is an interesting open question to understand whether our algorithms can be strengthened to work in this setting. Some positive evidence in this direction is due to Blanc, Lange, Malik, and Tan [BLMT22], who show that adaptive and non-adaptive adversaries have qualitatively similar power in many settings.

## 1.3 Related Work

Robust estimation is a classic subfield of Statistics, see, for example, the classic works [Tuk60, Hub64]. Our work fits more into the Computer Science literature on distribution estimation, initiated by the work of [KMR+94], which was in turn inspired by Valiant's PAC learning model [Val84]. Since then, several works have focused on algorithms for learning specific classes of distributions, see, e.g., [CDSS13, CDSS14a, CDSS14b, LS17, ABDH+20]. A recent line of work, initiated by [DKK+16, LRV16], focuses on computationally-efficient robust estimation of multivariate distributions, see, e.g., [DKK+17, SCV18, DKK+18, KSS18, HL18, DKK+19, LM21, LM22, BDJ+22, JKV23] and [DK22] for a reference. In contrast to all of these works, we focus on broad and generic connections between learnability and robust learnability, rather than studying robust learnability of a particular class of distributions.

Some of our algorithmic results employ tools from hypothesis selection, a problem which focuses on agnostic learning with respect to a specified finite set of distributions. The most popular approaches are based on ideas introduced by Yatracos [Yat85] and subsequently refined by Devroye and Lugosi [DL96, DL97, DL01]. Several others have studied hypothesis selection with an eye for several considerations, including running time, approximation factor, robustness, privacy, parallelization, and more [MS08, DDS12, DK14, SOAJ14, DKK+16, ABDH+18, AFJ+18, BKSW19, BKM19, GKK+20, BBK+22, AAK21].

Distribution learning under the constraint of differential privacy [DMNS06] has been an active area of research, see, e.g., [KV18, KLSU19, BS19, ASZ21, CWZ21, KMS+22b, KMS22a], and

---

[5]A natural open question is whether approximate DP learnability implies additively-robust approximate-DP learnability.

[6]And in the case of removals, much more natural

[KU20] for a survey. A number of these works have focused on connections between robustness and privacy [DL09, BKSW19, KSU20, AM20, BGS+21, LKKO21, LKO22, KMV22, RJC22, SM22, HKM22, GH22, HKMN23, AKT+23]. Again, these results either focus on specific classes of distributions, or give implications that require additional technical conditions, whereas we aim to give characterizations of robust learnability under minimal assumptions.

The question whether learnability under realizability assumptions extends to non-realizable setting has a long history. For binary classification tasks, both notions are characterized by the finiteness of the VC-dimension, and are therefore equivalent [VC71, VC74, BEHW89, Hau92]. [BPS09] show a similar result for online learning. Namely, that agnostic (non-realizable) learnability is characterized by the finiteness of the Littlestone dimension, and is therefore equivalent to realizable learnability.

Going beyond binary classification, recent work [HKLM22] shows that the equivalence of realizable and agnostic learnability extends across a wide variety of settings. These include models with no known characterization of learnability such as learning with arbitrary distributional assumptions and more general loss functions, as well as a host of other popular settings such as robust learning, partial learning, fair learning, and the statistical query model. This stands in contrast to our results for the distribution learning setting. We show that realizable learnability of a class of probability distributions does *not* imply its agnostic learnability. It is interesting and natural to explore the relationship between various notions of distribution learnability, which we have scratched the surface of in this work.

## 2   Learnability Implies Additive Robust Learnability

We recall Theorem 1.5, which shows that any class that is realizably learnable is also additive robustly learnable.

**Theorem 1.5.** *Any class of probability distributions $\mathcal{Q}$ which is realizably learnable, is also $\eta$-additively 2-robustly learnable for every $\eta \in (0, 1/4)$.*

We prove this theorem by providing an algorithm based on classical tools for hypothesis selection [Yat85, DL96, DL97, DL01]. These methods take as input a set of samples from an unknown distribution and a collection of hypotheses distributions. If the unknown distribution is close to one of the hypotheses, then, given enough samples, the algorithm will output a close hypothesis. Roughly speaking, our algorithm looks at all large subsets of the dataset, such that at least one will correspond to an uncontaminated set of samples. A learner for the realizable setting (whose existence we assumed) is applied to each to generate a set of hypotheses, and we then use hypothesis selection to pick one with sufficient accuracy. The proof of Theorem 1.7 (showing that subtractively robust learnability implies robust learnability) follows almost the exact same recipe, except the realizable learner is replaced with a learner robust to subtractive contaminations. We recall some preliminaries in Section A. We then prove Theorem 1.5 in Section B, and we formalize and prove a version of Theorem 1.7 in Section 2.1.

We note that $\alpha = 2$ and $\alpha = 3$ are often the optimal factors to expect in distribution learning settings, even for the case of finite distribution classes. For example, for proper agnostic learning the factor $\alpha = 3$ is known to be optimal for finite collections of distributions, which holds for classes with only 2 distributions [BKM19]. Similarly the factor of $\alpha = 2$ is optimal if the notion of learning is relaxed to improper learners [BKM19, CDSS14b]. While we are not aware of lower bounds for the additive setting, a small constant factor such as 2 is within expectations for these problems.

For the proof of Theorem 1.5, we refer the reader to Section B in the appendix.

### 2.1   Subtractive Robust Learnability Implies Robust Learnability

Similarly, we can show that robustness with respect to a subtractive adversary implies robustness with respect to a general adversary. We note that this theorem requires a change in constants from $\alpha$ to $(2\alpha + 4)$.

**Theorem 1.7.** *If a class $\mathcal{C}$ is $\eta$-subtractive $\alpha$-robustly learnable, then it is also $\eta$-$(2\alpha + 4)$-robustly learnable.*

The proof follows a similar argument as the proof of Theorem 1.5 and can be found in Section C in the appendix.

# 3  Learnability Does Not Imply Robust (Agnostic) Learnablity

In this section we show that there are classes of distributions which are realizably learnable, but not robustly learnable.

**Theorem 3.1.** *There are classes of distributions $\mathcal{Q}$, such that $\mathcal{Q}$ is realizably learnable, but for every $\alpha \in \mathbb{R}$, $\mathcal{Q}$ is not $\alpha$-robustly learnable. Moreover, the sample complexity of learning $\mathcal{Q}$ can be arbitrarily close to (but larger than) linear. Namely, for any super-linear function $g$, there is a class $\mathcal{Q}_g$, with*

- *$\mathcal{Q}_g$ is realizable learnable with sample complexity $n_{\mathcal{Q}_g}^{re}(\varepsilon, \delta) \leq \log(1/\delta)g(1/\varepsilon)$;*

- *for every $\alpha \in \mathbb{R}$, $\mathcal{Q}_g$ is* not *$\alpha$-robustly learnable.*

Note that this statement appears *slightly* weaker than Theorem 1.6, in that it holds for $\alpha$-robust learnability rather than $\eta$-subtractive $\alpha$-robust learnability. In fact, the two statements are incomparable, due to the order of quantifiers in the construction. Here we provide a single class which is not $\alpha$-robustly learnable for every $\alpha$, whereas in the proof of Theorem 1.6 we give a different class for each $\alpha$ (though the two constructions are similar). For simplicity we focus here on Theorem 3.1, whereas the proofs of Theorem 1.6 and other claims appear in Section D.

The key idea to the proof is to construct a class which is easy to learn in the realizable case, by having each distribution of the class have a unique support element that is not shared by any other distributions in the class. Distributions on which this "indicator element" has sufficient mass will be easily identified, independent of how rich the class is on other domain elements. That richness makes the class hard to learn from samples that miss those indicators. Furthermore, we construct the class in a way that its members are close in total variation distance to distributions that place no weight on those indicator elements.

This is done by making the mass on these indicator elements small, so that the members of a class of distributions that results from deleting these indicator bits are close to the initially constructed class, $\mathcal{Q}_g$. In order to make this work for every target accuracy and sample complexity, we need to have a union of such classes with decreasingly small mass on the indicator bits. In order for this to not interfere with the realizable learnability, we let the distributions with small mass on the indicator bits have most of their mass on one point $(0,0)$ that is the same for all distributions in the class. This ensures that distributions for which the indicator bit will likely not be observed because their mass is smaller than some $\eta$ are still easily $\varepsilon$-approximated by a constant distribution ($\delta_{(0,0)}$). Lastly we ensure the impossibility of agnostic learnability, by controlling the rate at which $\eta$ approaches zero to be faster than the rate at which $\varepsilon$ approaches zero. With this intuition in mind, we will now describe the construction and proof of this theorem.

*Proof.* We first define the distributions in $\mathcal{Q}_g$. Let $\{A_i \subset \mathbb{N} : i \in \mathbb{N}\}$ be an enumeration of all finite subsets of $\mathbb{N}$. Define distributions over $\mathbb{N} \times \mathbb{N}$ as follows:

$$q_{i,j,k} = \left(1 - \frac{1}{j}\right)\delta_{(0,0)} + \left(\frac{1}{j} - \frac{1}{k}\right)U_{A_i \times \{2j+1\}} + \frac{1}{k}\delta_{(i,2j+2)}, \tag{1}$$

where, for every finite set $W$, $U_W$ denotes the uniform distribution over $W$. For a monotone, super-linear function $g : \mathbb{N} \to \mathbb{N}$, we now let $\mathcal{Q}_g = \{q_{i,j,g(j)} : i, j \in \mathbb{N}\}$. The first bullet point of the theorem (the class is learnable) follows from Claim 3.2 and the second bullet point (the class is not robustly learnable) follows from Claim 3.3. $\qquad\square$

**Claim 3.2.** *For a monotone function $g : \mathbb{N} \to \mathbb{N}$, let $\mathcal{Q}_g = \{q_{i,j,g(j)} : i, j \in \mathbb{N}\}$. Then, the sample complexity of $\mathcal{Q}_g$ in the realizable case is upper bounded by*

$$n_{\mathcal{Q}_g}^{re}(\varepsilon, \delta) \leq \log(1/\delta)g(1/\varepsilon).$$

This claim can be proved by showing that the following learner defined by

$$\mathcal{A}(S) = \begin{cases} q_{i,j,g(j)} & \text{if } (i, 2j+2) \in S \\ \delta_{(0,0)} & \text{otherwise} \end{cases}$$

is a successful learner in the realizable case. Intuitively, this learner is successful for distributions $q_{i,j,g(j)}$ for which $j$ is large (i.e., $j > \frac{1}{\varepsilon}$), since this will mean that $d_{\mathrm{TV}}(q_{i,j,g(j)}, \delta_{(0,0)})$ is small. Furthermore, it is successful for distributions $q_{i,j,g(j)}$ for which $j$ is small (i.e., upper bounded by some constant dependent on $\varepsilon$), because this will lower bound the probability $1/g(j)$ of observing the indicator bit on $(i, 2j + 2)$. Once the indicator bit is observed the distribution will be uniquely identified.

**Claim 3.3.** *For every function $g \in \omega(n)$ the class $\mathcal{Q}_g$ is not $\alpha$-robustly learnable for any $\alpha > 0$.*

This claim can be proven by showing that for every $\alpha$, there is $\eta$, such that the class of distributions $Q'$ such that for every $q' \in Q'$ there is $q \in \mathcal{Q}_g$ with $d_{\mathrm{TV}}(q, q') < \eta$ which is not $\alpha\eta$-weakly learnable.[7] In particular, those for every $q' \in Q'$ there is $q \in \mathcal{Q}_g$ and $p$ such that $q = (1 - \eta)q' + \eta q$. We construct this class and show that it is not learnable by using the construction and Lemma 3 from [LBD23].

## 3.1 Existence of sample compression schemes

Sample compression schemes are combinatorial properties of classes that imply their learnability. For a variety of learning tasks, such as binary classification or expectation maximization a class has a sample compression scheme if and only if it is learnable [MY16, BHM+17]. For classification tasks, sample compression for realizable samples implies agnostic sample compression. [ABDH+20] used compression schemes to show learnability of classes of distributions in the realizable case, but left open the question if for learning probability distributions, the existence of realizable sample compression schemes implies the existence of similar schemes for the non-realizable (agnostic, or robust) settings. We provide a negative answer to this question.

More concretely, let $\mathcal{Q}$ be a class of distributions over some domain $X$. A compression scheme for $\mathcal{Q}$ involves two agents: an encoder and a decoder.

- The encoder knows a distribution $q$ and receives a sample $S$ generated by this distribution. The encoder picks a bounded size sub-sample and sends it, possibly with a few additional bits to the decoder.
- The decoder receives the message and uses an agreed upon decoding rule (that may depend on $\mathcal{Q}$ but not on $q$ or $S$) to constructs a distribution $p$ that is close to $q$.

Of course, there is some probability that the samples are not representative of the distribution $q$, in which case the compression scheme will fail. Thus, we only require that the decoding succeed with constant probability.

We say that a class $\mathcal{Q}$ has a sample compression scheme (realizable or robust) if for every accuracy parameter $\varepsilon > 0$, the minimal required size of the sample $S$, and upper bounds on the size of the sub-sample and number of additional bits in the encoder's message depend only of $\mathcal{Q}$ and $\varepsilon$ (and are independent of the sample generating $q$ and on the sample $S$).

A realizable compression scheme is required to handle only $q$'s in $\mathcal{Q}$ and output $p$ such that $d_{\mathrm{TV}}(p, q) \leq \varepsilon$, while a robust compression scheme should handle any $q$ but the decoder's output $p$ is only required to be $\min_{q \in \mathcal{Q}}\{d_{\mathrm{TV}}(p, q)\} + \varepsilon$ close to $q$.

**Theorem 3.4** (Formal version of Theorem 1.8). *For every $\alpha \in \mathbb{R}$, the existence of a realizable compression scheme, does not imply the existence of an $\alpha$-robust compression scheme. That is, there is a class $\mathcal{Q}$ that has a realizable compression scheme, but for every $\alpha \in \mathbb{R}$, $\mathcal{Q}$ does not have an $\alpha$-robust compression scheme.*

*Proof.* Consider the class $\mathcal{Q} = \mathcal{Q}_g$ from Section 3. We note that this class has a compression scheme of size 1. However, from [ABDH+18], we know that having a $\alpha$-robust compression scheme implies $\alpha$-agnostic learnability. We showed in Theorem 1.6 that for every $\alpha$ and for every superlinear function $g$, the class $\mathcal{Q}_g$ is not $\alpha$-agnostically learnable. It follows that $\mathcal{Q}_g$ does not have an $\alpha$-robust compression scheme. $\square$

---

[7]We provide a definition for $\varepsilon$-weak learnability as Definition D.2. We note that the definition we provide is what would usually be referred to as $(1/2 - \varepsilon)$-weak learnability in the supervised learning literature. For simplicity, because $\varepsilon$ is our parameter of interest, we reparameterized the definition to be more intuitive.

In Section E, we present a precise quantitative definition of sample compression schemes, as well as the proof that the class $\mathcal{Q}_g$ has a sample compression scheme of size 1.

## 4 Implications of Private Learnability

Qualitatively speaking, differentially private algorithms offer a form of "robustness" – the output distribution of a differentially private algorithm is insensitive to the change of a single point in its input sample. The relationship between privacy and notions of "robustness" has been studied under various settings, where it has been shown that robust algorithms can be made private and vice versa [DL09, GH22, HKMN23].

For distribution learning, we find that: (1) the requirement of approximate differentially private learnability also does not imply (general) robust learnability; and (2) the stronger requirement of *pure* differentially pirvate learnability does imply robust learnability.

**Definition 4.1** (Differential Privacy [DMNS06]). *Let $X$ be an input domain and $Y$ to be an output domain. A randomized algorithm $A : X^m \to Y$ is $(\varepsilon, \delta)$-differentially private (DP) if for every $x, x' \in X^n$ that differ in one entry,*

$$\mathbb{P}[A(x) \in B] \leq e^\varepsilon \cdot \mathbb{P}[A(x') \in B] + \delta \qquad \text{for all } B \subseteq Y.$$

*If $A$ is $(\varepsilon, \delta)$-DP for $\delta > 0$, we say it satisfies approximate DP. If it satisfies $(\varepsilon, 0)$-DP, we say it satisfies pure DP.*

**Definition 4.2** (DP learnable class). *We say that a class $\mathcal{C}$ of probability distributions is (approximate) DP learnable if there exists a randomized learner $A$ and a function $n_{\mathcal{C}} : (0, 1)^4 \to \mathbb{N}$, such that for every probability distribution $p \in \mathcal{C}$, and every $(\alpha, \beta, \varepsilon, \delta) \in (0, 1)^4$, for $n \geq n_{\mathcal{C}}(\alpha, \beta, \varepsilon, \delta)$*

  1. *$A$ is $(\varepsilon, \delta)$-DP; and*

  2. *The probability over samples $S$ of size $n$ drawn i.i.d. from the distribution $p$, as well as over the randomness of $A$ that*

     $$d_{\mathrm{TV}}(p, A(S)) \leq \alpha$$

     *is at least $1 - \beta$.*

*We say $\mathcal{C}$ is pure DP learnable if a learner $A$ can be found that satisfies $(\varepsilon, 0)$-DP, in which case the sample complexity function $n_{\mathcal{C}} : (0, 1)^3 \to \mathbb{N}$ does not take $\delta$ as a parameter.*

**Theorem 4.3** (Approximate DP learnability vs. robust learnability).

  1. *If a class $\mathcal{Q}$ is approximate DP learnable, then $\mathcal{Q}$ is $\eta$-additive 2-robustly learnable for any $\eta \in (0, 1/4)$.*

  2. *There exists an approximate DP learnable class $\mathcal{Q}$ that is not $\alpha$-robustly learnable for any $\alpha \geq 1$.*

Note that the first claim is immediate from Theorem 1.5, since approximate DP learnability implies learnability. To prove the second claim, we show that the learner for the class $\mathcal{Q}$ described in Theorem 3.1 can be made differentially private by employing stability-based histograms [BNS16]. The proof appears in Section F.

**Theorem 4.4** (Pure DP learnable vs. robustly learnable). *If a class $\mathcal{Q}$ is pure DP learnable, then $\mathcal{Q}$ is 3-robustly learnable.*

The proof relies on the finite cover characterization of pure DP learnability.

**Proposition 4.5** (Packing lower bound, Lemma 5.1 from [BKSW19]). *Let $\mathcal{C}$ be a class of distributions, and let $\alpha, \varepsilon > 0$. Suppose $\mathcal{P}_\alpha$ is a $\alpha$-packing of $\mathcal{C}$, that is, $\mathcal{P}_\alpha \subseteq \mathcal{C}$ such that for any $p \neq q \in \mathcal{P}_\alpha, d_{\mathrm{TV}}(p, q) > \alpha$.*

*Any $\varepsilon$-DP algorithm $A$ that takes $n$ i.i.d. samples $S$ from any $p \in \mathcal{C}$ and has $d_{\mathrm{TV}}(p, A(S)) \leq \alpha/2$ with probability $\geq 9/10$ requires*

$$n \geq \frac{\log |\mathcal{P}_\alpha| - \log \frac{10}{9}}{\varepsilon}.$$

*Proof of Theorem 4.4.* Let $\alpha, \beta > 0$. Pure DP learnability of $\mathcal{Q}$ implies that there exists a 1-DP algorithm $A_{DP}$ and $n = n_{\mathcal{C}}(\alpha/12, 1/10, 1)$ such that for any $p \in \mathcal{Q}$, with probability $\geq 9/10$ over the sampling of $n$ i.i.d. samples $S$ from $p$, as well as over the randomness of the algorithm $A_{DP}$, we have $d_{\mathrm{TV}}(p, A_{DP}(S)) \leq \alpha/12$. By Proposition 4.5, any $\alpha/6$-packing $\mathcal{P}_{\alpha/6}$ of $\mathcal{Q}$ has

$$|\mathcal{P}_{\alpha/6}| \leq \exp(m) \cdot (10/9).$$

Let $\widehat{\mathcal{Q}}$ be such a maximal $\alpha/6$-packing. By maximality, $\widehat{\mathcal{Q}}$ is also an $\alpha/6$-cover of $\mathcal{Q}$. Hence, running Yatracos' 3-robust finite class learner (Theorem A.1) $A$ over $\widehat{\mathcal{Q}}$ with

$$n_{\widehat{\mathcal{Q}}}(\alpha/2, \beta) = O\left(\frac{\log|\widehat{\mathcal{Q}}| + \log(1/\beta)}{(\alpha/2)^2}\right)$$

samples drawn i.i.d. from $p$ yields, with probability $\geq 1 - \beta$

$$
\begin{aligned}
d_{\mathrm{TV}}(p, A(S)) &\leq 3\min\{d_{\mathrm{TV}}(p, p') : p' \in \widehat{\mathcal{Q}}\} + \alpha/2 \\
&\leq 3(\min\{d_{\mathrm{TV}}(p, p') : p' \in \mathcal{Q}\} + \alpha/6) + \alpha/2 \\
&= 3\min\{d_{\mathrm{TV}}(p, p') : p' \in \mathcal{Q}\} + \alpha. \qquad \square
\end{aligned}
$$

Note that Yatracos' algorithm for hypothesis selection can be replaced with a pure DP algorithm for hypothesis selection (Theorem 27 of [AAK21]) in order to achieve the following stronger implication.

**Theorem 4.6** (Pure DP learnable vs. robustly learnable)**.** *If a class $\mathcal{Q}$ is pure DP learnable, then $\mathcal{Q}$ is pure DP 3-robustly learnable.*

## 5   Conclusions

We examine the connection between learnability and robust learnability for general classes of probability distributions. Our main findings are somewhat surprising in that, in contrast to most known leaning scenarios, learnability does *not* imply robust learnability. We also show that learnability *does* imply additively robust learnability. We use our proof techniques to draw new insights related to compression schemes and differentially private distribution learning.

## Acknowledgments

Thanks to Argyris Mouzakis for helpful conversations in the early stages of this work. AB was supported by an NSERC Discovery Grant and a David R. Cheriton Graduate Scholarship. GK was supported by a Canada CIFAR AI Chair, an NSERC Discovery Grant, and an unrestricted gift from Google. TL was supported by a Vector Research Grant and a Waterloo Apple PhD Fellowship in Data Science and Machine Learning.

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

## A  Additional Preliminaries

We recall a classic theorem for the problem of hypothesis selection. Given a set of candidate hypothesis distributions, the algorithm selects one which is close to an unknown distribution (to which we have sample access). The requisite number of samples from the unknown distribution is logarithmic in the size of the set of candidates.

**Theorem A.1** (Yatracos' 3-robust learner for finite classes (Theorem 4.4 of [ABDH+20], based on Theorem 1 of [Yat85]))**.** *Let $\mathcal{C}$ be a finite class of distributions over a domain $\mathcal{X}$. There exists an algorithm $A$ such that for any $(\alpha, \beta) \in (0, 1)^2$ and for any distribution $p$ over $\mathcal{X}$, given a sample $S$ of size*

$$m = O\left(\frac{\log|\mathcal{C}| + \log(1/\delta)}{\varepsilon^2}\right)$$

*drawn i.i.d. from $p$, we have that with probability $\geq 1 - \delta$,*

$$d_{\mathrm{TV}}(p, A(S)) \leq 3 \cdot \min\{d_{\mathrm{TV}}(p, p') : p' \in \mathcal{C}\} + \varepsilon.$$

We also use the following form of the Chernoff bound.

**Proposition A.2** (Chernoff bounds)**.** *Let $X = \sum_{i=1}^{m} X_i$ where $X_1 \dots X_m$ are independent draws from Bernoulli($p$). For any $t \in (0, 1)$, $\mathbb{P}[X/m \leq (1 - t)p] \leq \exp(-t^2 mp/2)$.*

## B  Learnability Implies Additive Robust Learnability

In this section, we provide the proof for Theorem 1.5.

**Theorem 1.5.** *Any class of probability distributions $\mathcal{Q}$ which is realizably learnable, is also $\eta$-additively 2-robustly learnable for every $\eta \in (0, 1/4)$.*

*Proof.* Recall that $\mathcal{Q}$ is a class of probability distributions which is realizably learnable. We let $\mathcal{A}_{\mathcal{Q}}^{re}$ be a realizable learner for $\mathcal{Q}$, with sample complexity $n_{\mathcal{Q}}^{re}$. Let accuracy parameters $\varepsilon, \delta > 0$ be arbitrary. We will define $n_1 \geq \max\left\{2n_{\mathcal{Q}}^{re}\left(\frac{\varepsilon}{9}, \frac{\delta}{5}\right), \frac{162(1+\log(\frac{5}{\delta}))}{\varepsilon^2}\right\}$ and $n_2 \geq \frac{162\left(2(\eta + \frac{2\varepsilon}{9})n_1 \log(n_1) + \log(\frac{5}{\delta})\right)}{\varepsilon^2} \geq \frac{162(1+\log(\frac{5}{\delta}))}{\varepsilon^2}$, and $n = n_1 + n_2$ be their sum.

Our additive robust learner will receive a sample $S \sim (\eta q + (1-\eta)p)^n$ of size $n$. We can view a subset $S' \subset S$ as the "clean" part, being i.i.d. generated by $p$. The size of this clean part $|S'| = n'$ is distributed according to a binomial distribution $\mathrm{Bin}(n, 1-\eta)$. By a Chernoff bound (Proposition A.2), we get

$$\mathbb{P}\left[n' \leq \left(1 - \frac{\varepsilon}{9} - \eta\right)n\right] \leq \mathbb{P}\left[n' \leq \left(1 - \frac{\varepsilon}{9} - \eta + \frac{\eta\varepsilon}{9}\right)n\right] = \mathbb{P}\left[n' \leq \left(1 - \frac{\varepsilon}{9}\right)(1-\eta)n\right]$$

$$\leq \exp\left(-\left(\frac{\varepsilon}{9}\right)^2 n(1-\eta)/2\right) = \exp\left(-\frac{\varepsilon^2}{162}n(1-\eta)\right) \leq \exp\left(-\frac{\varepsilon^2}{162}n\left(1 - \frac{3}{4}\right)\right)$$

Thus, given that $n = n_1 + n_2 \geq 2\left(\frac{162(1+\log(\frac{5}{\delta}))}{\varepsilon^2}\right)$ with probability at least $1 - \frac{\delta}{5}$, we have $n' \geq n\left(1 - \eta - \frac{\varepsilon}{9}\right)$. For the rest of the argument we will now assume that we have indeed $n' \geq n\left(1 - \eta - \frac{\varepsilon}{9}\right)$. The learner now randomly partitions the sample $S$ into $S_1$ and $S_2$ of sizes $n_1$ and $n_2$, respectively. Now let $S_1' = S_1 \cap S'$ and $S_2' = S_2 \cap S'$ be the intersections of these sets with the clean set $S'$, and $n_1'$ and $n_2'$ be their respective sizes. We note that $n_1' \sim \mathrm{Hypergeometric}(n, n', n_1)$ and $n_2' \sim \mathrm{Hypergeometric}(n, n', n_2)$.[8] Thus, assuming $m' \geq m(1 - \eta - \frac{\varepsilon}{9})$ using Proposition A.2, we have that

$$\Pr\left[|S_1'| \leq \left(1 - \eta - \frac{2\varepsilon}{9}\right)n_1\right] \leq e^{-\frac{2}{81}\varepsilon^2 n_1}$$

---

[8]Recall that $\mathrm{Hypergeometric}(N, K, n)$ is the random variable of the number of "successes" when $n$ draws are made without replacement from a set of size $N$, where $K$ elements of the set are considered to be successes.

and

$$\Pr\left[|S_2'| \leq \left(1 - \eta - \frac{2\varepsilon}{9}\right)n_2\right] \leq e^{-\frac{2}{81}\varepsilon^2 n_2},$$

where the probability is over the random partition of $S$. We note that by our choices of $n_1$ and $n_2$, with probability $1 - \frac{2\delta}{5}$, the clean fractions of $S_1$ and $S_2$ (namely $\frac{|S_1'|}{|S_1|}$ and $\frac{|S_2'|}{|S_2|}$) are each at least $\left(1 - \eta - \frac{2\varepsilon}{9}\right)$.

Let

$$\hat{\mathcal{H}} = \left\{\mathcal{A}_\mathcal{Q}^{re}(S'') : S'' \subset S_1 \text{ with } |S''| = \left(1 - \eta - \frac{2\varepsilon}{9}\right)n_1\right\}$$

be the set of distributions output by the realizable learning algorithm $\mathcal{A}_\mathcal{Q}^{re}$ when given as input all possible subsets of $S_1$ of size exactly $\left(1 - \eta - \frac{2\varepsilon}{9}\right)n_1$. For $\varepsilon < \frac{9\eta}{2}$, we know that this set of distributions $|\hat{\mathcal{H}}|$ is of size $\binom{n_1}{(1 - \eta - \frac{2\varepsilon}{9})n_1} \leq n_1^{2\eta n_1}$. By the guarantee of the realizable learning algorithm $\mathcal{A}_\mathcal{Q}^{re}$, if there exists a "clean" subset $S_1' \subset S_1$ where $|S_1'| \geq n_1(1 - \eta - \frac{2\varepsilon}{9})$ (i.e., $S_1' \sim p^{n_1(1-\eta-\frac{2\varepsilon}{9})}$), then with probability $1 - \frac{\delta}{5}$ there exists a candidate distribution $q^* \in \hat{\mathcal{H}}$ with $d_{\mathrm{TV}}(p, q^*) = \frac{\varepsilon}{9}$.

We now define and consider the *Yatracos sets*.[9] For every $q_i, q_j \in \hat{\mathcal{H}}$, define the Yatracos set between $q_i$ and $q_j$ to be $A_{i,j} = \{x : q_i(x) \geq q_j(x)\}$.[10] We let

$$\mathcal{Y}(\hat{\mathcal{H}}) = \{A_{i,j} \subset \mathcal{X} : q_i, q_j \in \hat{\mathcal{H}}\}$$

denote the set of all pairwise Yatracos sets between distributions in the set $\hat{\mathcal{H}}$.

We now consider the $A$-distance [KBG04] between two distributions with respect to the Yatracos sets, i.e., we consider

$$d_{\mathcal{Y}(\hat{\mathcal{H}})}(p', q') = \sup_{B \in \mathcal{Y}(\hat{\mathcal{H}})} |p'(B) - q'(B)|.$$

This distance looks at the supremum of the discrepancy between the distributions across the Yatracos sets. Consequently, for any two distributions $p', q'$ we have $d_{\mathrm{TV}}(p', q') \geq d_{\mathcal{Y}(\hat{\mathcal{H}})}(p', q')$, since total variation distance is the supremum of the discrepancy across *all* possible sets. Furthermore, if $q', p' \in \hat{\mathcal{H}}$, then $d_{\mathrm{TV}}(p', q') = d_{\mathcal{Y}(\hat{\mathcal{H}})}(p', q')$, since either of the Yatracos sets between the two distributions serves as a set that realizes the total variation distance between them.

Suppose there is some $q^* \in \hat{\mathcal{H}}$ with $d_{\mathrm{TV}}(p, q^*) \leq \frac{\varepsilon}{9}$. Then for every $q \in \hat{\mathcal{H}}$:

$$\begin{aligned}
d_{\mathrm{TV}}(p, q) &\leq d_{\mathrm{TV}}(p, q^*) + d_{\mathrm{TV}}(q^*, q) \\
&\leq \frac{\varepsilon}{9} + d_{\mathcal{Y}(\hat{\mathcal{H}})}(q^*, q) \\
&\leq \frac{\varepsilon}{9} + d_{\mathcal{Y}(\hat{\mathcal{H}})}(q^*, p) + d_{\mathcal{Y}(\hat{\mathcal{H}})}(p, q) \\
&\leq \frac{\varepsilon}{9} + d_{\mathrm{TV}}(q^*, p) + d_{\mathcal{Y}(\hat{\mathcal{H}})}(p, q) \\
&\leq \frac{2\varepsilon}{9} + d_{\mathcal{Y}(\hat{\mathcal{H}})}(p, q).
\end{aligned}$$

Lastly, we will argue that we can empirically approximate $d_{\mathcal{Y}(\hat{\mathcal{H}})}(p, q)$, which we can then use to select a hypothesis. We note that, since $\mathcal{Y}(\hat{\mathcal{H}})$ is a finite set of size $\leq \binom{n_1}{(1 - \eta - \frac{2\varepsilon}{9})n_1}^2 = \binom{n_1}{(\eta + \frac{2\varepsilon}{9})n_1}^2 \leq ((n_1^{(\eta+\frac{2\varepsilon}{9})}))^2 = n_1^{2(\eta+\frac{2\eta}{9})n_1}$, we have uniform convergence[11] with respect to

---

[9]These sets are also sometimes called Scheffé sets in the literature.

[10]Note that this definition is asymmetric: $A_{i,j} \neq A_{j,i}$.

[11]A collection of sets $\mathcal{W}$ has the *uniform convergence property* if for every $(\varepsilon, \delta) \in (0, 1)^2$ there is a number $m_\mathcal{W}(\varepsilon, \delta)$ such that for every probability distribution $p$, with probability $\geq (1 - \delta)$ over samples $S$ of size

$\mathcal{Y}(\hat{\mathcal{H}})$. Recall that, we assumed that there is "clean" subsample $S_2'' \subset S_2$, which is i.i.d. distributed according to $p$ and of size $(1 - \frac{2\varepsilon}{9} - \eta)n_1$. We also note that the clean samples $S_1$ and $S_2$ are drawn independently from each other. Thus with probability $1 - \frac{\delta}{5}$, $S_2''$ is $\frac{\varepsilon}{9}$-representative of $p$ with respect to $\mathcal{Y}(\hat{\mathcal{H}})$. For a sample $S_0$ and a set $B \subset \mathcal{X}$, let us denote $S_0(B) = \frac{|S_0 \cap B|}{|S_0|}$. Because of the $\frac{\varepsilon}{9}$-representativeness of $S_2''$, we have for every $B \in \mathcal{Y}(\hat{\mathcal{H}})$:

$$|p(B) - S_2''(B)| \leq \frac{\varepsilon}{9}.$$

Thus,

$$|p(B) - S_2(B)|$$
$$= \left| p(B) - \frac{|S_2 \cap B|}{|S_2|} \right|$$
$$\leq \max\left\{ \left| p(B) - \frac{|S_2'' \cap B|}{|S_2|} \right|, \left| p(B) - \frac{|S_2'' \cap B| + (\eta + \frac{2\varepsilon}{9})n_2}{|S_2|} \right| \right\}$$
$$\leq \max\left\{ \left| p(B) - \frac{S_2''(B)|S_2''|}{n_2} \right|, \left| p(B) - \frac{S_2''(B)|S_2''| + (\eta + \frac{2\varepsilon}{9})n_2}{n_2} \right| \right\}$$
$$\leq \max\left\{ \left| p(B) - \left(1 - \eta - \frac{2\varepsilon}{9}\right) S_2''(B) \right|, \left| p(B) - \frac{S_2''(B)\left(1 - \eta - \frac{2\varepsilon}{9}\right)n_2 + \left(\eta + \frac{2\varepsilon}{9}\right)n_2}{n_2} \right| \right\}$$
$$\leq \max\left\{ |p(B) - S_2''(B)| + \left| S_2''(B) - \left(1 - \eta - \frac{2\varepsilon}{9}\right) S_2''(B) \right|, \left| p(B) - \left( S_2''(B)\left(1 - \eta - \frac{2\varepsilon}{9}\right) + \left(\eta + \frac{2\varepsilon}{9}\right) \right) \right| \right\}$$
$$\leq \max\left\{ \frac{3\varepsilon}{9} + \eta, \left| p(B) - S_2''(B)\left(1 - \eta - \frac{2\varepsilon}{9}\right) - \left(\eta + \frac{2\varepsilon}{9}\right) \right| \right\}$$
$$\leq \max\left\{ \frac{3\varepsilon}{9} + \eta, \left| p(B) - S_2''(B) + S_2''(B)\left(\eta + \frac{2\varepsilon}{9}\right) - \left(\eta + \frac{2\varepsilon}{9}\right) \right| \right\}$$
$$\leq \max\left\{ \frac{3\varepsilon}{9} + \eta, |p(B) - S_2''(B)| + \left| S_2''(B)\left(\eta + \frac{2\varepsilon}{9}\right) - \left(\eta + \frac{2\varepsilon}{9}\right) \right| \right\}$$
$$\leq \max\left\{ \frac{3\varepsilon}{9} + \eta, \frac{\varepsilon}{9} + |S_2''(B) - 1|\left(\eta + \frac{2\varepsilon}{9}\right) \right\}$$
$$\leq \max\left\{ \frac{3\varepsilon}{9} + \eta, \frac{\varepsilon}{9} + \left(\eta + \frac{2\varepsilon}{9}\right) \right\}$$
$$\leq \frac{3\varepsilon}{9} + \eta$$

Let the empirical $A$-distance with respect to the Yatracos sets be defined by

$$d_{\mathcal{Y}(\hat{\mathcal{H}})}(q, S) = \sup_{B \in d_{\mathcal{Y}(\hat{\mathcal{H}})}} |q(B) - S(B)|.$$

---

$n > n_{\mathcal{W}}(\varepsilon, \delta)$ generated i.i.d. by $p$, a sample $S$ is $\varepsilon$-*representative* for $\mathcal{W}$ with respect to $p$. Namely, for every $A \in \mathcal{W}$, $\left| \frac{|A \cap S|}{|S|} - p(A) \right| \leq \varepsilon$. If $\mathcal{W}$ is finite then $n_{\mathcal{W}}(\varepsilon, \delta) \leq \frac{\log(|\mathcal{W}|) + \log(1/\delta)}{\varepsilon^2}$. For more details see Chapter 4 in [SB14].

Now if the learner outputs $\hat{q} \in \arg\min_{q \in \hat{\mathcal{H}}} d_{\mathcal{Y}(\hat{\mathcal{H}})}(q, S)$, then putting all of our guarantees together, we get that with probability $1 - \delta$

$$
\begin{aligned}
d_{\mathrm{TV}}(\hat{q}, p) &\leq \frac{2\varepsilon}{9} + d_{\mathcal{Y}(\hat{\mathcal{H}})}(\hat{q}, p) \\
&\leq \frac{5\varepsilon}{9} + \eta + d_{\mathcal{Y}(\hat{\mathcal{H}})}(\hat{q}, S_2) \\
&\leq \frac{5\varepsilon}{9} + \eta + d_{\mathcal{Y}(\hat{\mathcal{H}})}(q^*, S_2) \\
&\leq \frac{8\varepsilon}{9} + 2\eta + d_{\mathcal{Y}(\hat{\mathcal{H}})}(q^*, p) \\
&\leq \frac{8\varepsilon}{9} + 2\eta + \frac{\varepsilon}{9} \leq 2\eta + \varepsilon.
\end{aligned}
$$

$\square$

## C   Robust Learnability with Subtractive Contamination Implies Robust Learnability with General Contamination

In this section we will provide the proof for Theorem 1.7.

**Theorem 1.7.** *If a class $\mathcal{C}$ is $\eta$-subtractive $\alpha$-robustly learnable, then it is also $\eta$-$(2\alpha + 4)$-robustly learnable.*

*Proof.* Let $\mathcal{C}$ be a concept class that is $\eta$-subtractively $\alpha$-robust learnable. Then there exists a successful $\eta$-subtractive $\alpha$-robust learner $\mathcal{A}_{\mathcal{C}}^{sub}$ with sample complexity $n_{\mathcal{C}}^{sub}$ for the class $\mathcal{C}$. Let $\varepsilon < \frac{9\eta}{2}$ and $\delta$ and be arbitrary.

Let $n_1 \geq \max\left\{ 2n_{\mathcal{C}}^{sub}(\frac{\varepsilon}{9}, \frac{\delta}{5})/(1 - \eta - \frac{2\varepsilon}{9}), \frac{162(1+\log(\frac{5}{\delta}))}{\varepsilon^2} \right\}$ and $n_2 \geq \left\{ \frac{(4(\eta + \frac{2\varepsilon}{9})n_1 \log(n_1) + \log(\frac{5}{\delta})))}{\varepsilon^2}, \frac{162(1+\log(\frac{5}{\delta}))}{\varepsilon^2} \right\}$. Lastly let $n = n_1 + n_2$.

Let $p \in \mathcal{C}$ be arbitrary. The $\alpha$-$\eta$-robust learner receives a sample $S \sim p^m$ such that there is $q \in \mathcal{C}$ such that $d_{\mathrm{TV}}(p, q) = \eta$. Thus there exists a distributions $q_1, q_2, q_3$, such that $(1 - \eta)q_1 + \eta q_2 = p$ and $(1 - \eta)q_1 + q_3 = q$. We now use the same learning strategy as in Theorem 1.5: We split the sample randomly into two subsamples $S_1$ and $S_2$, where we use $S_1$ to learn candidate sets and then use $S_2$ to select the hypothesis from the candidate set. The goal in both settings is to find as close an approximation to $q_1$ as possible. The candidate based on $S_1$ is created by feeding subsamples of $S_1$ into the subtractively robust learner in such a way that with high probability one of the subsamples is guaranteed to be i.i.d. generated by $q_1$ and thus (with high probability) yield a good hypothesis. More precisely, the learner randomly splits the sample $S$ into $S_1$ and $S_2$ with $|S_1| = n_1$ and $|S_1| = n_2$. We now define the "clean" part of $S' \subset S$, i.e. the part of $S'$ that is i.i.d. distributed according to $q_1$. We note that the size of this "clean" sample $|S'| = n'$ is a random variable and distributed according to the binomial distributions $Binom(n, 1 - \eta)$. Now applying Chernoff bound, with the same argument as in the proof of Theorem 1.5, we get that with probability $1 - \frac{\delta}{5}$, we have $n' \geq n(1 - \eta - \frac{\varepsilon}{9})$. Now let $S_1' = S_1 \cap S'$ and $S_2' = S_2 \cap S'$ be the "clean parts" of the subsamples $S_1$ and $S_2$ respectively. The sizes $|S_1''| = n_1'$ and $|S_2''| = n_2''$ We note that $n_1' \sim Hypergeometric(n, n(1 - \eta), n_1)$ and $n_2' \sim Hypergeometric(n, n(1 - \eta), n_2)$. Thus,

$$
Pr_{\text{random split}}\left[ |S_1'| \leq \left(1 - \eta - \frac{2\varepsilon}{9}\right) n_1 \right] \leq e^{-\frac{2}{81}\varepsilon^2 n_1}
$$

and

$$
Pr_{\text{random split}}\left[ |S_2'| \leq \left(1 - \eta - \frac{2\varepsilon}{9}\right) n_2 \right] \leq e^{-\frac{2}{81}\varepsilon^2 n_2}.
$$

Taking together the guarantees on our random splits and the size of $n'$, we note that by our choices of $n_1$ and $n_2$ with probability $1 - \frac{3\delta}{5}$, the fractions of the parts that are i.i.d. generated by $q_1$ (namely $\frac{|S_1''|}{|S_1'|}$ and $\frac{|S_2''|}{|S_2'|}$) are at least $(1 - \eta - \frac{2\varepsilon}{9})$. Going forward we will assume that this is indeed the case.

Let

$$\hat{\mathcal{H}} = \left\{ \mathcal{A}_{\mathcal{Q}}^{sub}(\tilde{S}) : \tilde{S} \subset S_1' \text{ with } |\tilde{S}| = \left(1 - \eta - \frac{2\varepsilon}{9}\right)n_1 \right\}.$$

Using our assumption that $|S_1'| \geq (1-\eta-\frac{2\varepsilon}{9})n_1$, we know that there is $S_1'' \subset S_1'$ with $\mathcal{A}_{\mathcal{Q}}^{sub}(S_1'') \in \mathcal{H}'$. As $S_1'' \sim q_1^{(1-\eta-\frac{2\varepsilon}{9})n_1}$, by the learning guarantee of $\mathcal{A}_{\mathcal{Q}}^{sub}$ with probability $1 - \frac{\delta}{5}$, there is a candidate distribution $q^* \in \hat{\mathcal{H}}$ with $d_{\mathrm{TV}}(p, q^*) \leq d_{\mathrm{TV}}(p, q_1) + d_{\mathrm{TV}}(q_1, q^*) = \eta + (\alpha\eta + \frac{\varepsilon}{9}) = (\alpha + 1)\eta + \frac{\varepsilon}{9}$.

We now consider the Yatracos sets. For every $q_i, q_j \in \hat{\mathcal{H}}$, let $A_{i,j} = \{x : q_i(x) \geq q_j(x)\}$ and let

$$\mathcal{Y}(\hat{\mathcal{H}}) = \left\{ A_{ij} \subset \mathcal{X} : q_i, q_j \in \hat{\mathcal{H}} \right\}.$$

We now consider the $A$-distance [KBG04] between two distributions with respect to the Yatracos sets, i.e., we consider

$$d_{\mathcal{Y}(\hat{\mathcal{H}})}(p', q') = \sup_{B \in \mathcal{Y}(\hat{\mathcal{H}})} |p'(B) - q'(B)|.$$

We note, that for any two distributions $p', q'$ we have $d_{\mathrm{TV}}(p', q') \geq d_{\mathcal{Y}(\hat{\mathcal{H}})}(p', q')$. Furthermore, if $q', p' \in \hat{\mathcal{H}}$, then $d_{\mathrm{TV}}(p', q') = d_{\mathcal{Y}(\hat{\mathcal{H}})}(p', q')$. Assume there is $q^* \in \hat{\mathcal{H}}$ with $d_{\mathrm{TV}}(p, q^*) \leq (\alpha + 1)\eta + \frac{\varepsilon}{9}$, then for every $q \in \hat{\mathcal{H}}$:

$$\begin{aligned}
d_{\mathrm{TV}}(p, q) &\leq d_{\mathrm{TV}}(p, q^*) + d_{\mathrm{TV}}(q^*, q) \\
&\leq (\alpha + 1)\eta + \frac{\varepsilon}{9} + d_{\mathcal{Y}(\hat{\mathcal{H}})}(q^*, q) \\
&\leq (\alpha + 1)\eta + \frac{\varepsilon}{9} + d_{\mathcal{Y}(\hat{\mathcal{H}})}(q^*, p) + d_{\mathcal{Y}(\hat{\mathcal{H}})}(p, q) \\
&\leq (\alpha + 1)\eta + \frac{\varepsilon}{9} + d_{\mathrm{TV}}(q^*, p) + d_{\mathcal{Y}(\hat{\mathcal{H}})}(p, q) \\
&\leq 2(\alpha + 1)\eta + \frac{2\varepsilon}{9} + d_{\mathcal{Y}(\hat{\mathcal{H}})}(p, q).
\end{aligned}$$

Lastly, we will argue that we can empirically approximate $d_{\mathcal{Y}(\hat{\mathcal{H}})}(p, q)$, which we can then use to select a hypothesis. We note that, since $\mathcal{Y}(\hat{\mathcal{H}})$ is a finite set of size $|\mathcal{Y}(\hat{\mathcal{H}})| = \left(\binom{n_1}{(1-\eta-\frac{2\varepsilon}{9})n_1}\right)^2 \leq n_1^{2(\eta+\frac{2\varepsilon}{9})n_1}$, we have uniform convergence with respect to $\mathcal{Y}(\hat{\mathcal{H}})$. Recall that $S_2' \sim q_1^{n_2'}$ and by our previous assumption $n_2' \leq n_2\left(1 - \eta - \frac{2\varepsilon}{9}\right)$. Thus by our choice of $n_2$, with probability $1 - \frac{\delta}{5}$, there is $S_2'' \subset S_2' \subset S_2$ with $|S_2''| = \left(1 - \eta - \frac{2\varepsilon}{9}\right)n_2$ such that $S_2''$ is $\frac{\varepsilon}{9}$-representative of $q_1$ with respect to $\mathcal{Y}(\hat{\mathcal{H}})$. For a sample $S_0$ and a set $B \subset \mathcal{X}$, let us denote $S_0(B) = \frac{|S_0 \cap B|}{|S_0|}$. Because of the $\frac{\varepsilon}{9}$-representativeness of $S_2'$, we have for every $B \in \mathcal{Y}(\hat{\mathcal{H}})$:

$$|q_1(B) - S_2''(B)| \leq \frac{\varepsilon}{9}$$

Thus,

$$|q_1(B) - S_2(B)| \le |q_1(B) - S_2''(B)| + |S_2''(B) - S_2(B)|$$

$$\le \frac{\varepsilon}{9} + \left| \frac{|S_2 \cap B|}{|S_2|} - \frac{|S_2'' \cap B|}{|S_2''|} \right|$$

$$\le \frac{\varepsilon}{9} + \left| \frac{|S_2 \cap B|}{n_2} - \frac{|S_2'' \cap B|}{n_2(1 - \eta - \frac{2\varepsilon}{9})} \right|$$

$$= \frac{\varepsilon}{9} + \left| \frac{|S_2 \cap B|(1 - \eta - \frac{2\varepsilon}{\eta}) - |S_2'' \cap B|}{(1 - \eta - \frac{2\varepsilon}{9})n_2} \right|$$

$$\le \frac{\varepsilon}{9} + \max\{ \frac{|(|S_2'' \cap B| + (\eta + \frac{2\varepsilon}{9})n_2)(1 - \eta - \frac{2\varepsilon}{9}) - |S_2'' \cap B||}{(1 - \eta - \frac{2\varepsilon}{9})n_2},$$

$$\frac{|(|S_2'' \cap B|(1 - \eta - \frac{2\varepsilon}{9}) - |S_2'' \cap B||}{(1 - \eta - \frac{2\varepsilon}{9})n_2} \}$$

$$\le \frac{\varepsilon}{9}$$

$$+ \max\{ \frac{|(|S_2'' \cap B| + (\eta + \frac{2\varepsilon}{9})n_2)(1 - \eta - \frac{2\varepsilon}{9}) - ((1 - \eta - \frac{2\varepsilon}{9})|S_2'' \cap B| + (\eta + \frac{2\varepsilon}{9})|S_2'' \cap B|)|}{n_2(1 - \eta - \frac{2\varepsilon}{9})},$$

$$\frac{n_2(\eta + \frac{2\varepsilon}{9})}{n_2(1 - \eta - \frac{2\varepsilon}{9})} \}$$

$$\le \frac{\varepsilon}{9} + \max \left\{ \frac{|(\eta + \frac{2\varepsilon}{9})n_2(1 - \eta - \frac{2\varepsilon}{9}) - |S_2'' \cap B|(\eta + \frac{2\varepsilon}{9})|}{(1 - \eta - \frac{2\varepsilon}{9})n_2}, \frac{(\eta + \frac{2\varepsilon}{9})}{(1 - \eta - \frac{2\varepsilon}{9})} \right\}$$

$$\le \frac{\varepsilon}{9} + \max \left\{ \frac{|(\eta + \frac{2\varepsilon}{9})(n_2(1 - \eta - \frac{2\varepsilon}{9})|)}{(1 - \eta - \frac{2\varepsilon}{9})n_2}, \eta + \frac{2\varepsilon}{9} \right\}$$

$$\le \frac{\varepsilon}{9} + \eta + \frac{2\varepsilon}{9} \le \frac{3\varepsilon}{9} + \eta$$

Let us remember that the empirical $A$-distance with respect to the Yatracos is defined by

$$d_{\mathcal{Y}(\hat{\mathcal{H}})}(q, S) = \sup_{B \in \mathcal{Y}} |q(B) - S(B)| .$$

Now if the learner outputs $\hat{q} \in \arg\min_{q \in \hat{\mathcal{H}}} d_{\mathcal{Y}(\hat{\mathcal{H}})}(q, S_2)$, then putting all of our guarantees together, with probability $1 - \delta$ we get

$$d_{\mathrm{TV}}(\hat{q}, p) \le 2(\alpha + 1)\eta + \frac{2\varepsilon}{9} + d_{\mathcal{Y}}(\hat{q}, p)$$

$$\le 2(\alpha + 1)\eta + \frac{2\varepsilon}{9} + d_{\mathcal{Y}}(\hat{q}, q_1) + d_{\mathcal{Y}}(q_1, p)$$

$$\le 2(\alpha + 1)\eta + \frac{2\varepsilon}{9} + \eta + (\eta + \frac{3\eta}{9}) + d_{\mathcal{Y}}(\hat{q}, S_2)$$

$$\le 2(\alpha + 2)\eta + \frac{5\varepsilon}{9} + d_{\mathcal{Y}}(\hat{q}, S_2)$$

$$\le 2(\alpha + 2)\eta + \frac{5\varepsilon}{9} + d_{\mathcal{Y}}(q^*, S_2)$$

$$\le 2(\alpha + 2)\eta + \frac{5\varepsilon}{9} + (\eta + \frac{3\varepsilon}{9}) + d_{\mathcal{Y}}(q^*, q_1)$$

$$\le (2\alpha + 3)\eta + \frac{8\varepsilon}{9} + \eta + d_{\mathcal{Y}}(q^*, p)$$

$$\le (2\alpha + 4)\eta + \varepsilon + d_{\mathrm{TV}}(q^*, p).$$

$\square$

# D   Learnability Does Not Imply Robust Learnability

We start with an upper bound, showing that our class $\mathcal{Q}_g$ is realizably learnable.

**Claim 3.2.** *For a monotone function $g : \mathbb{N} \to \mathbb{N}$, let $\mathcal{Q}_g = \{q_{i,j,g(j)} : i, j \in \mathbb{N}\}$. Then, the sample complexity of $\mathcal{Q}_g$ in the realizable case is upper bounded by*

$$n_{\mathcal{Q}_g}^{re}(\varepsilon, \delta) \leq \log(1/\delta)g(1/\varepsilon).$$

*Proof.* Let the realizable learner $\mathcal{A}$ be

$$\mathcal{A}(S) = \begin{cases} q_{i,j,g(j)} & \text{if } (i, 2j+2) \in S \\ \delta_{(0,0)} & \text{otherwise} \end{cases}$$

Note that for all $\mathcal{Q}_g$-realizable samples this learner is well-defined. Furthermore, we note that in the realizable case, whenever $\mathcal{A}$ outputs a distribution different from $\delta_{(0,0)}$, then $\mathcal{A}(S)$ outputs the ground-truth distribution, i.e., the output has TV-distance 0 to the true distribution. Lastly, we note, that for an i.i.d. sample $S \sim q_{i,j,g(j)}^n$, we have the following upper bound for the learner identifying the correct distribution:

$$\mathbb{P}_{S \sim q_{i,j,g(j)}^n}[\mathcal{A}(S) = q_{(j)}] = \mathbb{P}_{S \sim q_{i,j,g(j)}^n}[(i, 2j+2) \in S] = 1 - (1 - 1/g(j))^n.$$

We note, that since $g$ is a monotone function, if $\varepsilon \leq \frac{1}{j}$, then $g(j) \leq g(\frac{1}{\varepsilon})$ and therefore,

$$(1 - 1/g(j))^n \leq (1 - 1/g(1/\varepsilon))^n.$$

Furthermore for $q_{i,j,g(j)}$, we have that $d_{\mathrm{TV}}(\delta_{(0,0)}, q_{i,j,g(j)}) = \frac{1}{j}$.

Putting these two observations together, we get

$$\mathbb{P}_{S \sim q_{i,j,g(j)}^n}[d_{\mathrm{TV}}(\mathbb{A}(S), q_{i,j,g(j)}) \geq \varepsilon] \leq \begin{cases} (1 - 1/g(1/\varepsilon))^n & \text{if } \frac{1}{j} \leq \varepsilon \\ 0 & \text{if } \frac{1}{j} > \varepsilon \end{cases}.$$

Thus, for every $q \in \mathcal{Q}_g$,

$$\mathbb{P}_{S \sim q^n}[d_{\mathrm{TV}}(\mathbb{A}(S), q) \geq \varepsilon] \leq (1 - 1/g(1/\varepsilon))^n \leq \exp\left(-\frac{n}{g(1/\varepsilon)}\right).$$

Letting the left-hand side equal the failure probability $\delta$ and solving for $n$, we get,

$$\log \delta \geq \frac{-n}{g(1/\varepsilon)}$$
$$\log(\delta)g(1/\varepsilon) \geq -n$$
$$n \geq -\log(\delta)g(1/\varepsilon) = \log(1/\delta)g(1/\varepsilon).$$

Thus, we have a sample complexity bound of

$$n_{\mathcal{Q}_g}(\varepsilon, \delta) \leq \log(1/\delta)g(1/\varepsilon).$$

$\square$

Now, we show a lower bound, that our class $\mathcal{Q}_g$ is *not* robustly learnable. Before we do that, we require a few more preliminaries. For a distribution class $\mathcal{Q}$ and a distribution $p$, let their total variation distance be defined by

$$d_{\mathrm{TV}}(p, \mathcal{Q}) = \inf_{q \in \mathcal{Q}} d_{\mathrm{TV}}(p, q).$$

We also use the following lemma from [LBD23].

**Lemma D.1** (Lemma 3 from [LBD23])**.** *Let $\mathcal{P}_{\eta,4k} = \{(1-\eta)\delta_{(0,0)} + \eta U_{A \times \{2j+1\}} : A \subset [4k]\}$ For $\mathcal{Q} = \mathcal{P}_{\eta,4k}$, we have $n_{\mathcal{Q}}^{re}(\frac{\eta}{8}, \frac{1}{7}) \geq k$.*

Finally, we recall the definition of weak learnability, which says that a distribution class is learnable only for some particular value of the accuracy parameter.

**Definition D.2.** *A class $\mathcal{Q}$ is $\varepsilon$-weakly learnable, if there is a learner $\mathcal{A}$ and a sample complexity function $n : (0,1) \to \mathbb{N}$, such that for ever $\delta \in (0,1)$ and every $p \in \mathcal{Q}$ and every $n \geq n(\delta)$,*

$$\mathbb{P}_{S \sim p^n}[d_{\mathrm{TV}}(\mathcal{A}(S), p) \leq \varepsilon] < \delta.$$

Learnability clearly implies $\varepsilon$-weak learnability for every $\varepsilon \in (0,1)$. While in some learning models (e.g., binary classification) learnability and weak learnability are equivalent, the same is not true for distribution learning [LBD23].

We are now ready to prove that $\mathcal{Q}_g$ is not robustly learnable.

**Claim 3.3.** *For every function $g \in \omega(n)$ the class $\mathcal{Q}_g$ is not $\alpha$-robustly learnable for any $\alpha > 0$.*

*Proof.* Consider

$$q'_{i,j} = \left(1 - \frac{1}{j}\right)\delta_{(0,0)} + \frac{1}{j}U_{A_i \times \{2j+1\}}$$

Note that $d_{\mathrm{TV}}(q'_{i,j}, \mathcal{Q}_g) = \frac{1}{g(j)}$. Therefore, in order to show that $\mathcal{Q}_g$ is not $\alpha$-robustly learnable, it is sufficient to show that there are $j$ and $\varepsilon$, such that the class $\mathcal{Q}'_j = \{q'_{i,j} : i \in \mathbb{N}\}$ is not $(\frac{\alpha}{g(j)} + \varepsilon)$-weakly learnable.

We will now show that for any $\gamma < \frac{1}{8j}$, the class $\mathcal{Q}'_j = \{q'_{i,j} : i \in \mathbb{N}\}$ is not $\gamma$-weakly learnable. Recalling notation from Lemma D.1, we note that that for every $n \in \mathbb{N}$ the class $P_{\frac{1}{j}, 4k} \subset \mathcal{Q}'_j$. By monotonicity of the sample complexity and Lemma D.1, we have $n_{\mathcal{Q}'_j}(\frac{1}{8j}, \frac{1}{7}) \geq n_{P_{1/j, 4n}}(\frac{1}{8j}, \frac{1}{7}) \geq n$, proving that this class is not weakly learnable.

Lastly, we need to show that for every $\alpha$, there are $\varepsilon$ and $j$, such that this claim holds for $\gamma \leq (\frac{\alpha}{g(j)} + \varepsilon)$. That is, we need to show that there are $\varepsilon$ and $j$, such that

$$\frac{\alpha}{g(j)} + \varepsilon < \frac{1}{8j}.$$

Let $\varepsilon = \frac{1}{16j}$. Now let $g$ be any superlinear function, i.e., for every $c \in \mathbb{R}$, there is $t_c \in \mathbb{N}$, such that for every $t \geq t_c$, $g(t) \geq ct$. This implies that, for any $\alpha \in \mathbb{R}$, there is $j \in \mathbb{N}$ such that $g(j) > 16j\alpha$. Thus for any super-linear function $g$ and any $\alpha \in \mathbb{R}$, the class $\mathcal{Q}_g$ is not $\alpha$-robustly learnable.

$\square$

## D.1 Proof of Theorem 1.6

The result of Theorem 1.6 follows directly from the construction of class $\mathcal{Q}_g$ for Theorem 3.1, the Claim 3.2 that shows this class is realizable learnable, and an adapted version for Claim 3.3, which states the following:

**Claim D.3.** *For every $\alpha$, there is $g(t) \in O(t^2)$, such that for every $0 \leq \eta \leq \frac{1}{16\alpha}$ the class $\mathcal{Q}_g$ is not $\eta$-subtractive $\alpha$-robustly learnable.*

*Proof.* Let $\alpha > 1$ be arbitrary. Let $g : \mathbb{N} \to \mathbb{N}$ be defined by $g(t) = 32\alpha t^2$ for all $t \in \mathbb{N}$. Now for every $0 \leq \eta \leq \frac{1}{16\alpha}$, there exists some $j$, such that $\frac{j}{g(j)}. \leq \eta \leq \frac{1}{16\alpha j}$

For such $j$, we consider the distributions

$$q'_{i,j} = \left(1 - \frac{1}{j}\right)\delta_{(0,0)} + \frac{1}{j}U_{A_i \times \{2j+1\}}$$

as in the proof of Claim 3.2. Recall that the element of $\mathcal{Q}_g$ are of the form

$$q_{i,j,g(j)} = \left(1 - \frac{1}{j}\right)\delta_{(0,0)} + \left(\frac{1}{j} - \frac{1}{g(j)}\right)U_{A_i \times \{2j+1\}} + \frac{1}{g(j)}\delta_{(i,2j+2)}$$

Then we have,

$$q_{i,j,g(j)} = \left(1 - \frac{1}{j}\right)\delta_{(0,0)} + \left(\frac{1}{j} - \frac{1}{g(j)}\right)U_{A_i \times \{2j+1\}} + \frac{1}{g(j)}\delta_{(i,2j+2)} =$$

$$= \left(1 - \frac{1}{j}\right)\delta_{(0,0)} + \left(\frac{g(j) - j}{jg(j)}\right)U_{A_i \times \{2j+1\}} + \frac{j}{jg(j)}\delta_{(i,2j+2)} =$$

$$= \left(\frac{g(j) - j}{g(j)}\right)\left(\left(1 - \frac{1}{j}\right)\delta_{(0,0)} + \frac{1}{j}U_{A_i \times \{(2,j+1)\}}\right) + \frac{j}{g(j)}\left(\left(1 - \frac{1}{j}\right)\delta_{(0,0)} + \frac{1}{j}\delta_{(i,2j+2)}\right) =$$

$$= \left(1 - \frac{j}{g(j)}\right)q'_{i,j} + \frac{j}{g(j)}\left(\left(1 - \frac{1}{j}\right)\delta_{(0,0)} + \frac{1}{j}\delta_{(i,2j+2)}\right).$$

Thus for every element $q'_{i,j}$ of the class $\mathcal{Q}'_j = \{q'_{i,j} : i \in \mathbb{N}\}$, there is a distribution $p$, such that $(1 - \eta)q'_{i,j} + \eta p \in \mathcal{Q}_g$. That is, every element of $\mathcal{Q}'_j$ results from the $\eta$-subtractive contamination of some element in $\mathcal{Q}_g$. Thus, for showing that $\mathcal{Q}_g$ is not $\eta$-subtractive $\alpha$-robustly learnable, it is sufficient to show, that $\mathcal{Q}'_j$ is not $(\alpha\eta + \varepsilon)$-weakly learnable for $\varepsilon = \frac{1}{16j}$. As we have seen in the proof of Claim 3.3, we can use Lemma D.1 to show that for every $n$, we have $n_{\mathcal{Q}'_j}(\frac{1}{8j}, \frac{1}{7}) \geq n$. Lastly, we need that $\frac{1}{8j} \geq \alpha\eta + \varepsilon$, or after replacing $\varepsilon$, we need $\frac{1}{16j} \geq \alpha\eta$. This follows directly from the choice of $g$.

$\square$

# E Existence of sample compression schemes

We adopt the [ABDH$^+$20] definition of sample compression schemes. We will let $\mathcal{C}$ be a class of distribution over some domain $\mathcal{X}$. A compression scheme for $\mathcal{C}$ involves two parties: an encoder and a decoder.

- The encoder has some distribution $q \in \mathcal{C}$ and receives $n$ samples from $q$. They send a succinct message (dependent on $q$) to the decoder, which will allow the decoder to output a distribution close to $q$. This message consists of a subset of size $\tau$ of the $n$ samples, as well as $t$ additional bits.
- The decoder receives the $\tau$ samples and $t$ bits and outputs a distribution which is close to $q$.

Since this process inherently involves randomness (of the samples drawn from $q$), we require that this interaction succeeds at outputting a distribution close to $q$ with only constant probability.

More formally, we have the following definitions for a decoder and a (robust) compression scheme.

**Definition E.1** (decoder, Definition 4.1 of [ABDH$^+$20]). *A decoder for $\mathcal{C}$ is a deterministic function $\mathcal{J} : \bigcup_{n=0}^{\infty} \mathcal{X}^n \times \bigcup_{n=0}^{\infty}\{0,1\}^n \to \mathcal{C}$, which takes a finite sequence of elements of $\mathcal{X}$ and a finite sequence of bits, and outputs a member of $\mathcal{C}$.*

The formal definition of a compression scheme follows.

**Definition E.2** (robust compression schemes, Definition 4.2 of [ABDH$^+$20]). *Let $\tau, t, n : (0,1) \to \mathbb{Z}_{\geq 0}$ be functions, and let $r \geq 0$. We say $\mathcal{C}$ admits $(\tau, t, n)$ $r$-robust compression if there exists a decoder $\mathcal{J}$ for $\mathcal{C}$ such that for any distribution $q \in \mathcal{C}$ and any distribution $p$ on $\mathcal{X}$ with $d_{\mathrm{TV}}(p,q) \leq r$, the following holds:*

*For any $\varepsilon \in (0,1)$, if a sample $S$ is drawn from $p^{n(\varepsilon)}$, then, with probability at least $2/3$, there exists a sequence $L$ of at most $\tau(\varepsilon)$ elements of $S$, and a sequence $B$ of at most $t(\varepsilon)$ bits, such that $d_{\mathrm{TV}}(\mathcal{J}(L, B), \mathcal{C}) \leq r + \varepsilon$.*

Note that $S$ and $L$ are sequences rather than sets, and can potentially contain repetitions.

**Theorem E.3** (Compression implies learning, Theorem 4.5 of [ABDH$^+$20]). *Suppose $\mathcal{C}$ admits $(\tau, t, n)$ $r$-robust compression. Let $\tau'(\varepsilon)\tau(\varepsilon) + t(\varepsilon)$. Then $\mathcal{C}$ can be $\max\{3, 2/r\}$-learned in the agnostic setting using*

$$O\left(n\left(\frac{\varepsilon}{6}\right)\log\left(\frac{1}{\delta}\right) + \frac{\tau'(\varepsilon/6)\log(n(\varepsilon/6)\log_3(1/\delta)) + \log(1/\delta)}{\varepsilon^2}\right) = \widetilde{O}\left(n\left(\frac{\varepsilon}{6}\right) + \frac{\tau'(\varepsilon/6)\log n(\varepsilon/6)}{\varepsilon^2}\right)$$

*samples. If $\mathcal{Q}$ admits $(\tau, t, n)$ non-robust compression, then $\mathcal{Q}$ can be learned in the realizable setting using the same number of samples.*

**Theorem E.4.** *The class $\mathcal{Q} = \mathcal{Q}_g$ from Section 3 has a compression scheme of message size 1 (i.e., using just a single sample point).*

*Proof.* Let $n(\varepsilon)$ be $10/g(\varepsilon)$ and, give a sample $S$ of at least that size, let the encoder pick a subset $L(S) \subseteq S$ be

$$L(S) = \begin{cases} \{(i, 2j + 2)\} & \text{if } (i, 2j + 2) \in S \\ \{(0, 0)\} & \text{otherwise} \end{cases}$$

Let the decoder output

$$\mathcal{J}(L) = \begin{cases} q_{i,j,g(j)} & \text{if } (i, 2j + 2) \in L \\ \delta_{(0,0)} & \text{otherwise} \end{cases}$$

With this construction established, the analysis follows very similarly to the analysis in the proof of Claim 3.2. $\qquad\square$

We note that the claim of Theorem 1.8 (and Theorem 3.4) follows directly.

## F  Approximate DP learnability vs robust learnability

We prove the second claim in Theorem 4.3 by showing that the learner for the class $\mathcal{Q}$ described in Theorem 3.1 can be made differentially private by employing stability-based histograms [KKMN09, BNS16].

**Proposition F.1** (Stability-based histograms [KKMN09, BNS16], Lemma 4.1 from [AAL21]). *Let $\mathcal{X}$ be a domain of examples. Let $K$ be a countable index set, and let $(h_k)_{k \in K}$ be a sequence of disjoint histogram bins over $\mathcal{X}$. For every $(\alpha, \beta, \varepsilon, \delta) \in (0, 1)^4$, there is an $(\varepsilon, \delta)$-DP algorithm that takes a dataset $S$ of size $n$ and with probability $\geq 1 - \beta$, outputs bin frequency estimates $(f_k)_{k \in K}$ such that*

$$\left| f_k - \frac{|\{x \in S : x \in h_k\}|}{n} \right| \leq \alpha$$

*for all $k \in K$, so long as*

$$n \geq \Omega \left( \frac{\log(1/\beta\delta)}{\alpha\varepsilon} \right).$$

*Proof of Theorem 4.3.* Recall the realizably-but-not-robustly learnable class of distributions $\mathcal{Q}_g = \{q_{i,j,g(j)} : i, j \in \mathbb{N}\}$ over $\mathbb{N}^2$ from Theorem 3.1, where $g : \mathbb{N} \to \mathbb{N}$ is a monotone, super-linear function and $q_{i,j,k}$ is as defined in (1).

Fix $(\alpha, \beta, \varepsilon, \delta) \in (0, 1)^4$. We define our DP learner $A_{DP}$ for $\mathcal{Q}_g$ as follows: we take a sample $S$ of size

$$n \geq \Omega \left( \frac{\log(1/(\beta/2)\delta)}{(1/4g(1/\alpha))\varepsilon} \right) + 32 \log(1/(\beta/2)) g(1/\alpha)$$

and run the stability-based histogram from Proposition F.1 targeting $(\varepsilon, \delta)$-DP, with singleton histogram buckets $(h_{(a,b)})_{(a,b) \in \mathbb{N}^2}$, each $h_{(a,b)} = \{(a, b)\}$. This yields frequency estimates $(f_{(a,b)})_{(a,b) \in \mathbb{N}^2}$ with $|f_{(a,b)} - |\{x \in S : x = (a,b)\}|| \leq 1/4g(1/\alpha)$ for all $(a, b) \in \mathbb{N}^2$ with probability $\geq 1 - \beta/2$. Then let

$$A_{DP}(S) = \begin{cases} q_{i,j,g(j)} & \text{if } f_{(i,2j+2)} \geq 1/2g(1/\alpha) \\ \delta_0 & \text{otherwise.} \end{cases}$$

Note that by post-processing $A_{DP}$ is indeed $(\varepsilon, \delta)$-DP. Now suppose $q_{i,j,g(j)}$ is our unknown distribution. There are two cases:

If $1/j \leq \alpha$, conditioned on the success of the histogram algorithm, the only possible outputs of $A_{DP}$ are $\delta_0$ and $q_{i,j,g(j)}$, so $d_{\mathrm{TV}}(A_{DP}(S), q_{i,j,g(j)}) \leq \alpha$ with probability $\geq 1 - \beta/2$.

If $1/j > \alpha$, we have $1/g(1/\alpha) < 1/g(j)$. By the second term in $n$ and Chernoff bounds (Proposition A.2), we can conclude that

$$\mathbb{P}\left[\frac{|\{x \in S : x = (i, 2j+2)\}|}{n} \leq \frac{3}{4g(1/\alpha)}\right] \leq \beta/2.$$

If the above event does not occur and if the histogram algorithm does not fail, $A_{DP}$ outputs $q_{i,j,g(j)}$ exactly. So $d_{\mathrm{TV}}(A_{DP}(S), q_{i,j,g(j)}) = 0$ with probability $\geq 1 - \beta$. $\qquad\square$

