# OpenReview forum: "Distribution Learnability and Robustness"
_NeurIPS.cc/2023/Conference — NeurIPS 2023 poster_

### Official Review · Reviewer_2DNr · 2023-06-19

**Soundness:** 4 excellent
**Presentation:** 2 fair
**Contribution:** 3 good
**Rating:** 6
**Confidence:** 3

**Summary:**

This paper studies the interaction between learnability and several notions of "robust" learnability, wherein candidate distributions can be corrupted in various ways.  The authors show an equivalence between learnability and a robust version of learnability in which a contaminated distribution can be added to the nominal distribution.  The authors study how differential privacy interacts with learning, showing that DP can imply learnability under certain contamination models, but not others.  Sample compression schemes are also explored.

**Overall assessment.**  This paper is strong on several fronts.  It considers a wide selection of problems related to distribution learnability.  And the technical quality of the paper is high.  On the negative side, I think the biggest drawback is the writing.  The clarity with which a paper like this is written is crucial, as the results are of a technical nature, thus requiring careful, clear explanations so that the reader can follow along.  In its current form, I think this paper falls short of the mark.  If the authors can demonstrably improve the writing and presentation, I will raise my score accordingly.

**Strengths:**

**Breadth of settings.**  This paper considers numerous problem settings spanning additive (and subtractive) contamination, differential privacy, learnability (in the sense of PAC learning), and sample compression.  One imagines that the scope of this paper could have been reduced, i.e., to the interaction of DP and learnability, and the paper would still constitute a solid contribution.  In an age when it can be advantageous to increase one's throughput by spreading content over multiple publications, I appreciated that the authors endeavored to give the reader the full picture capturing how learnability manifests on all of these different axes.

**Soundness.**  From what I can tell, the paper is quite sound.  The paper presents informal discussions and in some cases informal proofs in the main text.  The appendix is fully rigorous, with each main result proved in its own subsection.

**Weaknesses:**

**Motivation/Relevance to practice.**  There is little to no motivation for the problem under consideration.  The authors begin the introduction with a description of the problem of distribution learning without mentioning why one might be interested in this problem and/or what bearing this problem has on practical ML settings.  It's certainly worth noting that this is a more theoretical paper, but even still, I think that the paper would be stronger -- and would appeal to a wider audience -- if the authors motivated the problem in a more verbose way.  As written, this paper would only be accessible or feel relevant to those who are already familiar with why distribution learning is a well-studied problem, which is a narrow slice of the ML/statistics community.

**Definitions.**  I found the definitions on the first few pages hard to parse.  The three definitions contained in Def. 1.1 are all almost exactly the same; they each differ by at most a couple of words.  I think it would make much more sense to condense this by using some sort of meta-definition, and then listing the three definitions as special cases.  This also applies to Definition 1.4.  In the case of Definition 1.4, the only difference relative to Def. 1.1 seems to be that the learning algorithm is randomized, which could surely be summarized in a more concise way.

Another related point is that the authors state the definitions with relatively little context.  It would be help to illustrate each of these definitions with an example or a figure.  As I mentioned earlier, the differences between each of these definitions is relatively subtle, and it's asking a lot from the reader to keep all seven definitions on pages 2-3 in their heads at once, along with the fine distinctions that separate them.

**Writing.**  Much of the prose is notably spare in a way that makes the paper more challenging to read.  Results (e.g., Theorem 2.1) are often without any preamble.  It would be worth thinking about reorganizing the paper so that each result is accompanied by an intuitive explanation of the relevance of the result and how it fits into the broader context of the paper.  While I like the attempt to informally present the theorems before offering a more rigorous treatment after Section 1, I don't think that it works here.  This is in part because it asks the reader to consider each result at three levels of abstraction.  First, the results are stated informally in Section 1.  Then, in the remainder of the main text, the same results are stated formally, but the proofs are informal or sketched.  Finally, the full treatment is given in the appendix.  I think that asking the reader to jump around at these different levels of abstraction is, in this case, confusing.  Perhaps the authors could shorten the intro and stick to the results as stated from Section 2 onwards?

Relatedly, the intro seems too long here.  The reader has to wait until page 5 to learn what the authors feel the contributions are.  This dilutes the standard linear flow of ideas that characterizes papers that are easy to follow.  To make the paper easier to follow, I would recommend rewriting the introduction so that (a) the problem of distribution learning is well-motivated, (b) the main results are discussed, but not stated, and (c) the contributions are listed by -- at the very latest -- the end of the second page.

**How is this "robust?"**  One question that jumped out to me was as follows: How are the definitions in Definition 1.1 "robust?"  I.e., what are they robust against?  The authors claim that the name "robust learnability" is due to the fact that they are accounting for the ground truth distribution $p$ may not be an element of the class $\mathcal{C}$.  Would it not be more accurate to call these definitions "agnostic" rather than "robust?"  I can see how one would regard the corruption-based definitions in Def. 1.2 and Def. 1.3 as robust, because the distribution $p$ is mixed with a corrupted distribution.  However, in the sense of Definition 1.1, I don't understand the use of the word "robust."

**Miscellaneous.**

* It looks like a some points the authors write $(\alpha,\eta)-robustly learnable and $\alpha-\eta$-robustly learnable (c.f. Def. 1.1(3) vis-a-vis Def 1.4(3)).  It seems like it would be good to pick on notation and stick with it.
* Why is the TV distance chosen here?  The authors assert that this distance metric makes sense, but they give no argument, leaving the reader with some confusion.
* It would be worth adding a discussion section after the main results but before the conclusion.  After reading through the results, it would be useful if the authors could piece them together in the broader context of distribution learnability.  The authors could also discuss how these results could lead to future research on this problem, and/or whether this could lead to new algorithms that would be of practical use.


**Questions:**

See above.

---

> ### Author Rebuttal · Authors · 2023-08-09
>
> Thank you for the detailed and positive review! We are glad to hear the reviewer appreciates the breadth and technical quality of the results we present. We acknowledge the fair criticisms made regarding writing, and appreciate the many concrete suggestions offered by the reviewer to remedy these issues. We will take them into account in future revisions of our manuscript.
>
> In the following, we address specific concerns raised by the reviewer.
>
> —------------------------------------
>
> **On presentation:**
>
> The main critique of the paper is related to the presentation and organization of the paper, rather than the results themselves. These comments are well-received, and we are happy to take them into account as best as we can.
>
> Indeed, stating the main results (even informally) within the first two pages is an excellent suggestion, and I think we can do that to communicate the story early on. We thank the reviewer for giving many concrete suggestions. Given the stringent page limit and the thoroughness of our set of results, it may be difficult to fit all the suggestions into the body, but we will try our best, and include a “full version” in the supplement which is maximally readable.
>
> **On motivations:**
>
> Distribution learning is a classical problem in both statistics and computer science, which has seen significant study at NeurIPS and related conferences (see [1,2,3,4,5], note that [5] received a best paper award at NeurIPS 2018). We will emphasize the tradition and relevance of this work to the community.
>
> Learning in TV distance is perhaps the most common and established problem in distribution learning, we will add discussion and references to this, as well as pointers to learning in other distances.
>
> **On the terminology of robust:**
>
> Indeed, robustness and agnostic learning are closely related, and we generally use the terms interchangeably. Consider first the agnostic setting from the CS literature, traditionally defined as follows: there exists some underlying class of distributions, but an adversary instead selects an arbitrary distribution from within a TV ball of the class to draw samples from. This is more frequently used to capture model misspecification, but could also be used to capture adversarial contamination of the distribution.
>
> This is our non-adaptive setting (Definitions 1.2 and 1.3), though we note that this is also called the Huber contamination model in the robust statistics subfield of statistics. We thus emphasize that both “agnostic” and “robust” have been used to refer to this setting, depending on the community (where our paper is at their intersection). On the other hand, the adaptive version in Definition 1.4 is almost exclusively referred to in the algorithmic statistical estimation literature (i.e., [6]). Due to the overlapping terminology, we choose to use agnostic and robust interchangeably for all definitions, and differentiate between Definitions 1.2/1.3 and 1.4 using the terms adaptive and non-adaptive. We will clarify that we use these two terms interchangeably and why in the final version of the paper.
>
> **On the various definitions of robust learning:**
>
> One criticism is that the definitions are not clear or too similar, making it hard to parse. We will put more effort into explaining the different aspects of the definition in the main text of the paper.
>
> We also like the idea of a meta-definition and will take it into consideration. There are basically 3 types of adversaries: one who can add data, one who can remove data, and one who can do both. Now those adversaries either act on the distribution, giving rise to Definition 1.1, 1.2 and 1.3 respectively or they act on the sample, giving rise to Definition 1.4. While it is easy to state these intuitively, being formal is a bit more cumbersome, leading to the somewhat-clunky definitions. We will try to add an informal discussion to clarify, as well as add a table with these two axes to highlight the distinctions and give a clearer picture of the definitions.
>
> —--------------------------------
>
> We hope that the proposed changes will help make the paper more accessible. We are looking forward to a fruitful discussion with the reviewers about how to best get our message across and appreciate the reviewer’s feedback.
>
> **References**
>
> [1] Ishaq Aden-Ali, Hassan Ashtiani, and Christopher Liaw. Privately Learning Mixtures of Axis-Aligned Gaussians. NeurIPS’21.
>
> [2] Ilias Diakonikolas, Elena Grigorescu, Jerry Li, Abhiram Natarajan, Krzysztof Onak, and Ludwig Schmidt. Communication-Efficient Distributed Learning of Discrete Distributions. NIPS’17.
>
> [3] Ananda Theertha Suresh, Alon Orlitsky, Jayadev Acharya, and Ashkan Jafarpour. Near-optimal-sample estimators for spherical gaussian mixtures. NIPS’14.
>
> [4] Siu-On Chan, Ilias Diakonikolas, Rocco A. Servedio, Xiaorui Sun. Near-Optimal Density Estimation in Near-Linear Time Using Variable-Width Histograms NIPS’14.
>
> [5] Hassan Ashtiani, Shai Ben-David, Nicholas Harvey, Christopher Liaw, Abbas Mehrabian, Yaniv Plan. Nearly tight sample complexity bounds for learning mixtures of Gaussians via sample compression schemes. NeurIPS’18.
>
> [6] Ilias Diakonikolas, Gautam Kamath, Daniel Kane, Jerry Li, Ankur Moitra, and Alistair Stewart. Robust Estimators in High Dimensions without the Computational Intractability. STOC’16.

---

> > ### Comment · Reviewer_2DNr · 2023-08-14
> > **Rebuttal response**
> >
> > Thank you for your rebuttal response.  It's a shame that one cannot upload a new PDF, because I would quite like to read another draft of this paper, given that the updated bits all center around the presentation.  However, in the spirit of looking for reasons to accept rather than reasons to reject, I'll trust that the authors will follow up on the motivation, definitions, and writing style in the camera-ready version of their paper.  In particular, I think that adding motivation and historical context for this problem would be helpful, particularly for someone like your reviewer, who (as you can tell based on my review) is only loosely familiar with the problem being studied here (hence, `Confidence=3`).  Given that these changes will -- in my view -- improve the paper, it's only fair that I increase my score.

---

### Official Review · Reviewer_1ajk · 2023-07-06

**Soundness:** 3 good
**Presentation:** 4 excellent
**Contribution:** 3 good
**Rating:** 6
**Confidence:** 3

**Summary:**

This paper studies distribution learnability, with a focus on the relationship between the realizable and the agnostic/robust settings. This relationship is also studied in the context of sample compression schemes and differential privacy.

In particular, the authors show separation results: learnability in the realizable setting implies learnability when an adversary can modify the training data by adding training samples. They show however that this does not hold when the adversary can remove part of training data. In fact the authors identify that an adversary having subtractive power implies robust learnability in the more general case where data can be replaced. In terms of connection to sample compression schemes, the main result is that the existence of sample compression schemes in the realizable setting does not guarantee the existence of robust sample compression schemes. Finally, they show that pure DP learnability implies pure DP learnability when the adversary can replace data, but if one considers approximate DP, the implication breaks for adversaries that can subtract data to the training set (but not for adversaries that can add data).


**Strengths:**

- Good presentation, well written, big picture is given for most proofs and the paper easy to follow. The introduction is particularly thorough
- The impact is quite broad, with applications to sample compression schemes and differential privacy
- I believe these results would be of interest to the learning theory community
- The technical contributions seems appropriate for NeurIPS, given the significance of the conceptual contributions

Disclaimer: I am not very familiar with DP, so my assessment for the contributions is mainly based on Sections 1-3.


**Weaknesses:**

- Constant $\alpha$ (Thm 1.5): do you have a sense whether $\alpha=4$ the best one could hope for? This high constant seems to be a limitation of the work, in the sense that the only guarantee is that we are roughly at multiplicative factor of 4 away from the optimal distribution in the class $\mathcal{C}$. It's a bit difficult to get a sense of the significance and the text could use some clarification.

General comments/things to fix:
- p.2: "total variance distance" -> "total variation distance" (maybe include the mathematical definition too? No need for a new definition environment, just $d_{TV}(p,q)=...$)
- Section 6.6 in the appendix does not have any text below it


**Questions:**

- Terminology: robust and agnostic learnability are used interchangeably (e.g., l.32), which is in contrast to other set ups (e.g. binary classification), for which the two terms have vastly different meanings. The set up described on l.32-37 (and Defs 1.2 and 1.3) sounds more like the agnostic setting in spirit, where we want to compare the algorithm's output with the best possible one from the class $\mathcal{C}$. In my mind, robustness implies the presence of an adversary which modifies data (at training or testing time), so more in line with definition 1.4. I think using robust and agnostic interchangeably is a bit confusing and I would appreciate some clarification on the matter (here and on a future version of the paper), perhaps a distinction like suggested above.

General comments:
- Perhaps provide references for poisoning attacks on l.28, as it is particularly related to "adaptative robustness" (though this is in the context of classification).



**Limitations:**

Yes.

---

> ### Author Rebuttal · Authors · 2023-08-10
>
> Thank you for the kind review! We are glad the reviewer believes that our results would be of interest to the learning theory community – highlighting the breadth of settings considered and the significant conceptual contribution.
>
> In the following, we address specific points raised by the reviewer.
>
> —---------------------
>
> Regarding the constant $\alpha =4$ in Theorem 2.1 (6.3 in the supplementaries):
>
> We thank the reviewer for this question. We carefully examined our proof again and were able to improve the bound to $\alpha=2$. We did this by tightening the upper bound (page 16 of the supplementaries) on $|p(B) -S_2(B)|$ to $\frac{2\epsilon}{9} + \eta$, instead of$ \frac{3\epsilon}{9} + 2\eta$.
>
> This was possible by bounding the second part of the max by using triangle inequality in the following way:
> \begin{align*}
> |p(B) - ( S_2''(B)(1-\eta - \tfrac{\epsilon}{9}) + (\eta + \tfrac{\epsilon}{9}) )| &\leq |p(B) - S_2''(B) + S_2''(B)(\eta+\tfrac{\epsilon}{9}) - (\eta + \tfrac{\epsilon}{9}) | \\\\
> &\leq |p(B) - S_2''(B)| + |S_2''(B)(\eta+\tfrac{\epsilon}{9}) - (\eta + \tfrac{\epsilon}{9}) | \\\\
> &\leq \tfrac{\epsilon}{9} +|S_2’’(B)-1| (\eta + \tfrac{\epsilon}{9}) \leq \tfrac{2\epsilon}{9} + \eta.
> \end{align*}
>
>
> The last inequality holds  as $0 \leq S_2’’(B) \leq 1$.
>
> Furthermore, we note that $\alpha=2$ and $\alpha=3$ are often the optimal factors to expect in distribution learning settings, even for the case of finite distribution classes. For example, for proper agnostic learning the factor $\alpha=3$ is known to be optimal for finite collections of distributions  (see [1], which holds for classes with only 2 distributions). Similarly the factor of $\alpha=2$ is optimal if the notion of learning is relaxed to improper learners (see [2], also [1]). While we aren’t aware of lower bounds for the additive adaptive setting,  a multiplicative constant factor such as 2 is within expectations for these problems.
>
> **Regarding the terminology of robust vs. agnostic:**
>
> As we mention in Definition 1.1 we use $\alpha$-robust learning and $\alpha$-agnostic learning interchangeably. The “robustness” we are considering is with respect to corruptions to the training data, and is referred to as such in work on algorithmic statistical estimation [3], as opposed to test-time corruptions as considered in the adversarially robust learning literature [4]. The agnostic setting in distribution learning indeed corresponds to robustness with respect to corruptions in training data; as agnostic learning is defined as follows: there exists some underlying class of distributions, but an adversary instead selects an arbitrary distribution from within a TV ball of the class to draw samples from. This is more frequently used to capture model misspecification, but could also be used to capture adversarial contamination of the distribution.
>
> This is our non-adaptive setting (Definitions 1.2 and 1.3), though we note that this is also called the Huber contamination model in the robust statistics area of statistics. We thus emphasize that both “agnostic” and “robust” have been used to refer to this setting, depending on the community (where our paper is at their intersection). On the other hand, the adaptive version in Definition 1.4 is almost exclusively referred to in the algorithmic statistical estimation literature. Due to the overlapping terminology, we choose to use agnostic and robust interchangeably for all definitions, and differentiate between Definitions 1.2/1.3 and 1.4 using the terms adaptive and non-adaptive. We will clarify that we use these two terms interchangeably and why in the final version of the paper.
>
> Thank you for the idea of including references to poisoning attacks. We will do so in the final version.
>
> —--------------------------------
>
> Thanks also for pointing out some typos. We'll address those in the revised manuscript.
>
> If you have any additional concerns, questions, or suggestions, we would be happy to discuss further in the discussion phase.
>
> **References**
>
> [1] Olivier Bousquet, Daniel Kane, Shay Moran. The Optimal Approximation Factor in Density Estimation. COLT’19.
>
> [2] Siu-On Chan, Ilias Diakonikolas, Rocco A. Servedio, Xiaorui Sun. Near-Optimal Density Estimation in Near-Linear Time Using Variable-Width Histograms NIPS’14.
>
> [3] Ilias Diakonikolas, Gautam Kamath, Daniel Kane, Jerry Li, Ankur Moitra, and Alistair Stewart. Robust Estimators in High Dimensions without the Computational Intractability. STOC’16.
>
> [4] Omar Montasser, Steve Hanneke, and Nathan Srebro. VC Classes are Adversarially Robustly Learnable, but Only Improperly. COLT’19.

---

> > ### Comment · Reviewer_1ajk · 2023-08-11
> > **Response**
> >
> > Thanks for the response.
> >
> > *Update on the $\alpha$ term*: nice! If you end up updating the constant in the final version, could you include a discussion in line with the one in your response?
> >
> > *Robust vs agnostic terminology*: thank you for the clarification!
> >
> > I am looking forward to reading the above updates in the final version of your work.

---

### Official Review · Reviewer_Xv71 · 2023-07-06

**Soundness:** 3 good
**Presentation:** 4 excellent
**Contribution:** 3 good
**Rating:** 7
**Confidence:** 5

**Summary:**

This paper study the connection between learnability and robust (agnostic) learnability for general classes of distributions in the framework of distribution learning. Differ from the PAC learnablity, they show that in the distribution learning setting the learnability does not always implies agnostic learnablity. They also explore related implications in the context of compression schemes and differentially private learnability.

**Strengths:**

* This paper is well organized and easy to follow, I enjoy reading it.
* The result that learnability does not imply robust learnability is somewhat surpring.
* The proofs are solid, by my judgement.

**Weaknesses:**

N/A

**Questions:**

* I am a little confused about the definitions of adaptivity. In Definition 1.4 (1) and (3), what distribution does the new $\eta m$ examples follow? If they follow $p,$ these definitions seem to make no sence. If these examples are drawn from some unkown distriburtion, how should I understand the definition of **adaptive robust** learnablity?

* I would be very appreciated it if the authors can give some intuition about the contradiction between their findings and the results presented in [1].

Reference:
[1] Max Hopkins et al. Realizable learning is all you need. Colt 2022.

---

> ### Author Rebuttal · Authors · 2023-08-09
>
> Thank you for the kind review! Regarding your questions:
>
> **On the definitions of adaptivity:**
>
> In Definition 1.4 (1), the sample $S'$ of size $\eta m$ is arbitrary. Similarly, the sample $S’$ in (3) of the same definition is obtained by arbitrarily replacing a fixed fraction of instances in the sample with other arbitrary instances.
> Indeed, in both cases, one could more accurately replace “samples $S'$” with “arbitrary dataset $S’$”. We do not make any distributional assumptions on these manipulations (they do not arise from i.i.d. samples). Thus the adversary in this model is stronger than the one in definitions 1.1., 1.2 and 1.3, where we do make distributional assumptions on the added/replaced sample points.
>
> **On *Realizable Learning is All You Need*:**
>
> The biggest difference between our setting and that of [1] is that we consider learning classes of distributions, whereas [1] considers learning classes of functions. In general, these two settings are incomparable, and thus there is no contradiction.
>
> The main idea of [1] is to learn a finite “cover” $\mathcal H'$ (i.e., a set of hypotheses that is guaranteed to contain an epsilon-approximation for each hypothesis of the class) of possible labelings by the predefined hypothesis class $\mathcal H$ on an unlabelled sample $S_U$; this is possible since there is a only a finite set of possible labelings. They then use ERM on a labeled sample $S_L$ and the class $\mathcal H’$ to get a classifier with an agnostic learning guarantee.
>
> We note that in the distribution learning setting, cover-learning is not possible, since there is no analogue of “a sample being consistent with a hypothesis”: there can be an infinite number of possible generating distributions of $S_U$ in the class (that cannot be covered by a finite set of distributions). Basically, we give an example of a class which has no finite epsilon-cover (in terms of TV distance), but realizable learning is still possible (using mass on points that uniquely defines a distribution the the class).
>
> —--------------
>
> If you have any additional concerns, questions, or clarifications -- we would be happy to discuss further in the discussion phase.

---

> > ### Comment · Reviewer_Xv71 · 2023-08-17
> > **Response**
> >
> > Thanks for the response.

---

### Decision · Program_Chairs · 2023-09-21

**Decision:**

Accept (poster)

**Comment:**

This paper explores the relationship between learnability and robust learnability in the unsupervised setting of learning classes of probability distributions. The focus is on sample complexity bounds. It is shown that if one considers only additive corruptions, learnability implies robust learnability. But this is not the case with subtractive corruptions. Overall, the reviewers agreed that this is an interesting contribution that merits publication at NeurIPS.